# Assessing the Effect of the Magnitude of Spillovers on Global Supply Chains Using Quantile Vector Autoregressive and Wavelet Approaches

**Haibo Wang** [1] , **Lutfu Sagbansua** [2,*] and **Jaime Ortiz** [3]

1   Division of International Business and Technology Studies, Texas A&M International University, Laredo, TX 78041, USA
2   Department of Management and Marketing, Southern University and A&M College, Baton Rouge, LA 70807, USA
3   Department of International Business and Entrepreneurship, The University of Texas Rio Grande Valley, Edinburg, TX 78520, USA; jaime.ortiz@utrgv.edu
*   Correspondence: lutfu.sagbansua@subr.edu

**Abstract:** Overwhelmed by the negative impacts of the COVID-19 pandemic, global supply chains are being restructured and improved worldwide. It then becomes essential to accurately assess their vulnerabilities to external shocks and understand the relationships between key influential factors to obtain the desired results. This study provides a new conceptual econometric framework to examine the relationships between the purchasing managers' index, service purchasing managers' index, world equity index, unemployment rate, food and beverage historical prices, Baltic Dry Index, West Texas Intermediate Index, and carbon emissions. A quantile vector autoregressive (QVAR) model is used to assess the dynamic connectedness among Brazil, Russia, India, China, South Africa, and the United States based on such factors. A wavelet method is also utilized to assess the coherence between the time series. The results of the correlation and dynamic connectedness analyses for these countries reveal that the service purchasing managers' index offers the highest spillover value toward the other factors.

**Keywords:** supply chain disruption; QVAR; wavelet





## 1. Introduction

The pre-existing supply chain challenges that were magnified by the COVID-19 pandemic continue to create pressure at the most global scale. Disruptions of manufacturing operations resulting from curfews and regulations keep slowing the flow of goods and raw materials. However, the COVID-19 pandemic has not necessarily created any new challenges for supply chains. A survey conducted by Ernst & Young on senior-level supply chain executives underscored their plans to increase investment in supply chain technologies, such as artificial intelligence, business analytics, and robotic process automation, as an attempt to create collaborative, resilient, and sustainable supply chains. The survey revealed that only 2% of the respondents were fully prepared for the disruptions caused by external shocks. On the bright side, these disruptions have forced business executives to prioritize their supply chain issues and invest in building technical capabilities.

Although many industries were deeply affected by the COVID-19 pandemic, some of them were hit particularly hard. Shortages of semiconductors, batteries, fabrics, and critical elements exposed vulnerabilities in the auto manufacturing, electronic good, and textile industries. Record-low inventories in many industries are still holding back business activities and causing disruptions in industrial supply chains. One of the most significant impacts of such shortages is on price increases. Between April 2020 and April 2021, the prices of commodities tracked using the producer price index rose by seventeen percent,

while the change in commodity prices reached twenty-one percent between April 2021 and April 2022 [1]. To overcome such challenges, major companies in these industries have already started building more sustainable and resilient supply chains by geographically diversifying their supply sources, relying more on local suppliers, and investing in technological solutions throughout. Meanwhile, it is essential to start with identifying vulnerabilities before redesigning supply chains as there are many tradeoffs associated with potential solutions where supply source diversification is misaligned with efficiencies within the very same supply sources [2]. A recent analysis revealed the four top strategies being utilized by companies to mitigate the impact of COVID-19 on their supply chains. These are strengthening existing relationships, pursuing multiple and regionally diverse suppliers, relying on digital supply chain tools for increased visibility into their supply chain, and moving away from the just-in-time methodology to the just-in-case methodology [3].

Research efforts aiming to overcome these vulnerabilities currently focus on restructuring global value chains, improving demand prediction, and increasing transparency. Swanson and Santamaria conducted a bibliometric analysis to summarize the COVID-19 pandemic-related supply chain literature and reported that 84% of it was produced within the first ten months [4]. Another interesting finding is that although the pre-COVID-19 pandemic-related supply chain literature focused on the medical supply chain, the post-COVID-19 pandemic-related supply chain literature covered a wider range of industries using empirical methods, such as modeling and simulation. In an attempt to identify the main research streams, influential contributors, and disruption management strategies related to the supply chain performance in pandemic settings, Moosavi et al. presented bibliometric, network, and thematic analyses [5]. They found that technologies, such as artificial intelligence, the internet of things, and blockchains, as well as food and medical supply chains were the main research topics driven by the COVID-19 pandemic. Meanwhile, Vlachos reviewed 259 publications on supply chains and provided analyses of the impacts at the firm, industry, and economic sector levels, as well as a classification of supply chain resilience strategies [6].

Our paper is motivated by the fact that one of the most significant and difficult challenges that governments, policymakers, and societies have been facing recently is the assessment of the vulnerability of global supply chains to external shocks and building resilient supply chains to minimize negative impacts on their performance. Our research question focuses on the correlation between and spillover effects of factors, such as the purchasing managers' index (PMI), service purchasing managers' index (SPMI), unemployment rate (UR), world equity index (WEI), carbon emission futures (CEF), food and beverage historical prices (FBH), Baltic Dry Index (BDI), and West Texas Intermediate Index (WTI), before and after the COVID-19 pandemic to examine their impacts on global supply chains. Specifically, we pose the following research question:

Research Question: What is the connection between the factors that expose the vulnerability of global supply chains and the impact their performance? To assess the vulnerability and spillover effect, two quantitative models are implemented to evaluate the connectedness relationship and spillover effects of shocks on time-series data.

This paper makes the following novel contributions. First, we propose a new conceptual framework of dynamic connectedness analysis for time-series data from multiple countries with key indicators of supply chain performance. Second, a quantile vector autoregressive (QVAR) model reveals the dynamic connectedness among countries before and after the start of the pandemic. One important finding is that this study reveals the strong co-movement in the global supply chain during the period that is investigated. The results reveal high time-frequency dependence and a causal effect between the factors and global supply chains. Another result is that the co-movement between the unemployment and other factors is weaker compared to the co-movement between the PMI and SPMI, particularly up to 2017.

The rest of the paper is organized as follows. Section 2 reviews the supply chain research during the COVID-19 pandemic era. The theoretical econometric fundamentals are presented in Section 3, followed by the results and findings in Section 4. We provide conclusions in Section 5.

## 2. Literature Review

Our literature research on the impacts of external shocks on supply chains and overall economic activity focuses on two main streams. These are research on the impact of external shocks on supply chains and research on the effectiveness of various actions taken to respond to mitigate the impact of the external shocks.

### 2.1. Impact on Supply Chains

To evaluate the expectations for the evolution of supply chains in terms of geographic regions and industries, Vurdu examined the emergence of global value chains and the interdependence among countries [7]. Meyer et al. focused on the implications of the COVID-19 pandemic for supply chain constructs related to sustainability, resilience, and risk, using text mining [8]. Sarkis identified emerging consumer, organizational, policy, and supply chain behaviors by looking at the environmental sustainability of supply chains in a post-COVID-19 pandemic environment [9]. Carvalho et al. used an equilibrium model of production networks that took into account the macroeconomic disruption caused by the 2011 earthquake in East Japan along the supply chains [10]. Del Rio-Chanona et al. analyzed the constraints exerted on the U.S. economy after allowing shocks in its aggregate supply and aggregate demand to predict their impacts on factors, such as GDP, employment, and wages [11]. During the COVID-19 pandemic, Bekaert et al. studied output and price fluctuations, using real-time data on GDP growth and inflation, by modeling aggregate supply and aggregate demand shocks [12]. Around the same time, Brinca et al. measured labor supply and demand shocks at the sectoral level by estimating a Bayesian structural VAR model, attributing the drop in the rate of labor growth to supply effects [13]. Chen et al. studied the dynamic impact of the COVID-19 pandemic on consumption, using daily transaction data to reveal the sensitivity of demand to the pandemic severity [14]. Guerrieri et al. investigated whether supply shocks can lead to demand-deficient recessions and discussed the combination of monetary and fiscal policies [15]. Inoue and Todo used an agent-based model to simulate the impact of a complete Tokyo shutdown on the production losses in other prefectures and argued that the negative impact of the shutdown would rapidly propagate throughout because of supply and demand shortages [16]. Pichler et al. designed an economic model to address the features of the COVID-19 pandemic, including the inventory dynamics and feedback between the consumption and unemployment, and analyzed how shocks propagated through the production network [17]. Chetty et al. analyzed the heterogeneity of the impact of the COVID-19 pandemic across income levels, using weekly statistics on employment rates, job postings, consumer spending, and business revenues [18]. Khalfaoui et al. provided an empirical study on the roles of panic and stress related to the COVID-19 pandemic on green bond market volatilities [19]. Chen and Tillmann used a set of economic activity indicators, such as $NO_2$ emissions, maritime container trade, and mobility, to estimate the magnitude of lockdown spillovers [20]. Qian and Qiu examined the impact of political risk on corporate international supply chains and concluded that political risk decreased their number of purchases from foreign suppliers [21].

### 2.2. Responding to External Shocks

Freeman and Baldwin focused on the effects of the supply chain contagion of national lockdown measures on manufacturing industries [22]. Hyun et al. examined how global connectedness and market power affected the supply chain resilience and performance in response to the COVID-19 pandemic, using global stock market data, and concluded that higher global connectedness of supply chains led to more resilience to domestic shocks [23].

Bonadio et al. used a multi-industry quantitative framework covering 64 countries to investigate the impact that global supply chains had on GDP growth during the COVID-19 pandemic and argued that the nationalization of supply chains did not necessarily contribute to their resiliency because of an increasing dependency on domestic inputs, which were also disrupted by the lockdowns. [24] Using international trade variables, Heidary presented a system dynamic model to simulate the impact of the COVID-19 pandemic on the global supply chain in various scenarios and concluded that higher levels of flexibility in production capacity were an important strategy to cope with such disruptions [25]. Diaz Pacheco and Benedito investigated the responses of manufacturing and service businesses during the COVID-19 pandemic, using a qualitative multiple case study, and reported that supply chains did adapt activities, such as product design and development, budgeting, human resources, and logistics [26].

To estimate the impacts of COVID-19 as well as government responses on e-commerce sales, Han et al. utilized city–day panel data to illustrate the digital resilience of e-commerce during the pandemic and identified the logistics capacity as a key operational driver [27]. Blom et al. developed an optimization model to maximize the audience in a theater while satisfying the limitations imposed by the governments during the pandemic [28]. Li et al. used a two-tier supply chain to investigate the impact of government subsidy schemes and the channel power structure on the level of innovation in the supply chain [29]. The study provided guidance for governments on how to design effective subsidy schemes to improve innovation, investment, as well as social welfare, which are particularly important during challenging times. Zhai et al. also considered a two-tier supply chain composed of a retailer and a manufacturer to investigate service investment and pricing decisions under various power structures in the presence of demand disruptions [30].

Although the literature on the effects of the COVID-19 pandemic on supply chains has started to build sharply, there are a lack of quantitative models for examining the relationships between key factors mainly owing to the limitations on data availability. This research fills this gap to enable decisionmakers to implement informative decisions based on objective analytical results. Furthermore, the extant literature on the interrelations among factors do not account for the spillover effects of external shocks. Our paper contributes to the existing literature by assessing the connectedness of time-series data.

## 3. Methodology

To assess the connectedness of the time-series data, QVAR, and wavelet coherence methods can reveal the dynamic connectedness between variables and assess the spillover effects of shocks from a spatiotemporal perspective. Different quantiles in QVAR simulate diverse market conditions, while the wavelet coherence method simulates co-movement relationships.

### 3.1. Quantile Vector Autoregressive (QVAR) Method

To examine the connectedness among PMI, SPMI, UR, WEI, CEF, FBH, BDI, and WTI, we used the QVAR method. The workflow involving the time-series data is provided in Figure 1.

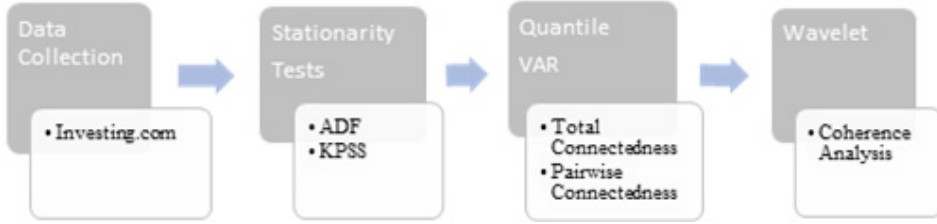

**Figure 1.** Workflow of the analysis.

The cointegration between multiple variables was tested using the Jarque–Bera (JB) goodness-of-fit test to evaluate the distribution of residuals of the VAR model for the

variables. The test provides skewness and kurtosis computations to determine normality. The residuals must be randomly distributed for normality, and *p*-values < 0.05 indicate a bidirectional cointegration. Pairwise cointegration between two variables implies the existence of system-wide cointegration.

$$
\begin{bmatrix} BDI_{it} \\ WTI_{it} \\ PMI_{it} \\ SPMI_{it} \\ WEI_{it} \\ CEF_{it} \\ FBH_{it} \\ UR_{it} \end{bmatrix} = A_0 + A_1 \begin{bmatrix} BDI_{it-1} \\ WTI_{it-1} \\ PMI_{it-1} \\ SPMI_{it-1} \\ WEI_{it} \\ CEF_{it} \\ FBH_{it} \\ UR_{it} \end{bmatrix} + A_2 \begin{bmatrix} BDI_{it-2} \\ WTI_{it-2} \\ PMI_{it-2} \\ SPMI_{it-2} \\ WEI_{it} \\ CEF_{it} \\ FBH_{it} \\ UR_{it} \end{bmatrix} + \cdots + A_p \begin{bmatrix} BDI_{it-p} \\ WTI_{it-p} \\ PMI_{it-p} \\ SPMI_{it-p} \\ WEI_{it} \\ CEF_{it} \\ FBH_{it} \\ UR_{it} \end{bmatrix} + CD_t + u_{it} \tag{1}
$$

where

$BDI_{it}$: Baltic Dry Index;
$WTI_{it}$: WTI;
$PMI_{it}$: purchasing managers' index;
$SPMI_{it}$: service purchasing managers' index;
$WEI_{it}$: world equity index;
$CEF_{it}$: carbon emission futures;
$FBH_{it}$: food and beverage historical prices;
$UR_{it}$: unemployment rate;

$$ i = \{1,..,6\}, and \ t = \{1,\ldots,7\}. $$

The augmented Dickey–Fuller (ADF) and Kwiatkowski–Phillips–Schmidt–Shin (KPSS) tests were used to confirm the stationarity of the time series [31,32]. The ADF test for time series $Y_t$ is given by a linear regression model as follows:

$$ \Delta_{Y_t} = \alpha + \beta t + \gamma Y_{t-1} + \theta_1 \Delta_{Y_{t-1}} + \ldots + \theta_{l-1} \Delta_{Y_{t-l+1}} + \epsilon_t \tag{2} $$

where $\alpha$ is a constant, $\beta$ is the coefficient of the time trend in time series $Y_t$, $l$ is the lag order of the autoregressive process, and $\epsilon_t$ is the stationary error of the linear regression model. The root test of the ADF for the null hypothesis, $\gamma = 0$, is given as follows:

$$ ADF_\gamma = \frac{\hat{\gamma}}{SE(\hat{\gamma})} \tag{3} $$

The KPSS test for time series $Y_t$ is given by the following linear regression using three parameters:

$$ Y_t = r_t + \beta_t + \epsilon_t \tag{4} $$

where $\alpha$ is a constant, $\beta_t$ is the deterministic trend, $r_t$ is the random walk $r_t = \alpha + \beta t + \sum_{i \leq t} u_i$, and $\epsilon_t$ is the stationary error of the linear regression model.

The QVAR for dynamic connectedness analysis [33] was used to assess the connectedness among countries. It cast the standard VAR models with a quantile regression model to measure the dynamics of the selected quantiles. Unlike the rolling regression model that sets the rolling window size arbitrarily, the QVAR defines multiple quantiles to avoid any loss of information. Furthermore, the QVAR is less sensitive to outliers because it uses a recursive information set rather than the standard VAR with constant parameters.

QVAR method for time series $Y_t$:

A recursive information set, $\Omega$, is used to model multiple quantiles [33]. For the lagged values of $\widetilde{Y}_t$ and the contemporaneous value of $\widetilde{Y}_{1,t+1}$, $\Omega_{1t} = \left\{ \widetilde{Y}_t, \widetilde{Y}_{t-1}, \cdots \right\}$ and $\Omega_{it} = \left\{ \widetilde{Y}_{i-1,t+1}, \Omega_{i-1,t} \right\}$.

For $p$ multiple distinct quantiles, $0 < \theta_1 < \theta_2 < \cdots < \theta_p < 1$, the QVAR model is given as follows:

$$Y_{t+1} = \omega + A_0 Y_{t+1} + A_1 Y_{t+1} + \epsilon_{t+1}, \; P\left(\epsilon_{i,t+1}^{\theta_j} < 0 \middle| \Omega_{1t}\right) = \theta_j, \; P = 1, \cdots, n, j = 1, \cdots, p \quad (5)$$

The error term, $\epsilon_{t+1}$, is quantile-specific and satisfies the condition of $P\left(\epsilon_{i,t+1}^{\theta_j} < 0 \middle| \Omega_{1t}\right) = \theta_j$. $Y_t = 1_p \otimes \widetilde{Y}_t$, where $1_p$ is a $p$-vector of ones. The matrices $A_0 = I_p \otimes \widetilde{A}_0$ have lower-triangular submatrices that have zeros on the main diagonal, and $A_1 = I_p \otimes \widetilde{A}_1$ to avoid trivial multicollinearity problems.

The forecast model of the QVAR can be described as the branches of a tree. For $p$ multiple distinct quantiles, the starting node, $\widetilde{Y}_{1,t+1}$, has $p$ branches ($p$ quantiles). At the end of each branch, there are $p$ more branches for $\widetilde{Y}_{2,t+1}$ and so on. The general form of the QVAR is given as follows:

$$Y_{t+1} = \omega A^0 Y_{t+1} + A^1 Y_t \epsilon_{t+1} \quad (6)$$

where $A^0 = \begin{bmatrix} A_0 & 0 & 0 \cdots & 0 \\ 0 & & & \\ \vdots & & \ddots & \vdots \\ 0 & & \cdots & 0 \end{bmatrix}$, and $A^1 = \begin{bmatrix} A_1 & A_2 & \cdots & A_q \\ I_{np} & 0 & \cdots & 0 \\ \vdots & \vdots & \ddots & \vdots \\ 0 & \cdots & I_{np} & 0 \end{bmatrix}$.

The QVAR model provides the natural environment for measuring impulse responses to a given shock by defining a set of future tail quantiles-of-interest and predicting the outcomes of variables that are conditional on the chosen shock. The structural quantile-impulse response function is given in terms of structural shocks as follows:

$$Y_t = (I_{np} - A_0)^{-1} \omega + (I_{np} - A_0)^{-1} A_1 Y_{t-1} + (I_{np} - A_0)^{-1} \epsilon_t \quad (7)$$

*3.2. Wavelet Method*

Originating from Fourier analysis [34,35], the wavelet method is used to assess the dynamics of co-movement among time series [19]. Xu, Liu, and Ortiz [36] used wavelet analysis to examine the amplitude and time-frequency distributions between the expected and actual inflation in the U.S. Grinsted, Moore, and Jevrejeva [37] applied the wavelet coherence and cross-wavelet transform to analyze the relationship in time-frequency space between two time series as follows:

$$D_{X_t Y_t}(r, p) = S\left(D_{X_t}^*(r, p) D_{Y_t}(r, p)\right) \quad (8)$$

where $D_{X_t}(r, p)$ and $D_{Y_t}(r, p)$ represent the continuous wavelet transforms of $X_t$ and $Y_t$ at scales $r$ and positions $p$, respectively. The superscript * is the complex conjugate, and $S$ is a smoothing operator in time and scale. Thus, the conference of time-series $X_t$ and $Y_t$ is given as follows:

$$\frac{\left| S\left(D_{X_t}^*(r, p) D_{Y_t}(r, p)\right) \right|^2}{S\left(|D_{X_t}(r, p)|^2\right) . S\left(|D_{Y_t}(r, p)|^2\right)} \quad (9)$$

## 4. Experimental and Empirical Findings

*4.1. Data*

Table 1 summarizes the econometric indicators used to measure the economic activities.

**Table 1.** Economic indicators.

| Indicator | Symbol | Explanation |
|---|---|---|
| Baltic Dry Index | BDI | Provides a benchmark for the price of transporting major raw materials by water |
| West Texas Intermediate | WTI | A crude oil sourced primarily from inland Texas |
| Purchasing Managers' Index | PMI | Represents the prevailing direction of economic trends in the manufacturing and service industries as viewed by purchasing managers |
| Service Purchasing Managers' Index | SPMI | Consists of a diffusion index calculated based on responses to a question asked to a panel of service sector providers about changes in the volume of business activity compared with the previous month |
| World Equity Index | WEI | Captures large- and mid-cap representations across 23 developed market (DM) countries and covers nearly 85% of the free float-adjusted market capitalization in each country (MSCI) |
| Carbon Emission Futures | CEF | Used as an indicator of industrial activity levels |
| Food and Beverage Historical Prices | FBH | Used as an indicator of food prices |
| Unemployment Rate | UR | Measures the percentage of the total workforce that is not working, yet actively seeking employment |

The BDI is used as an indicator that offers a clear perspective of the global demand for commodities and raw materials. WTI, as one of the highest-quality oils, serves as the main global oil benchmark and is included in the model as an indicator of industrial production. The PMI and SPMI are also included as indicators of the growth or expansion of the manufacturing and service sectors, respectively. The WEI is used as a measure of the equity market performance of emerging and developed markets. The CEF is added to the model as a proxy for the industrial output, while the FBH approximates the price patterns in critical goods during times of crisis. Finally, the UR reflects the degree of labor utilization in the job market. To perform the cointegration, causality, and connectedness analyses, time-series data were collected from various sources for four countries. Brazil, Russia, India, China, and South Africa have the largest emerging economies, while the U.S.A. has the key global market. The BRICS countries have undoubtedly become world economic powers; they are attempting to harness the forces of globalization to strengthen their international standing across multilateral settings [38]. The BRICS countries and the U.S. represent almost seventy-five percent of the combined worldwide trade (which goes hand-in-hand with supply chains) and enjoy a strategic political position in terms of their regional influence and, therefore, possess generality for assessing the effect of the magnitude of spillovers on global supply chains. The BDI, PMI, SPMI, UR, WEI, CEF, and FBH data were obtained from https://www.investing.com (accessed on 1 August 2023), while the WTI data were obtained from the Federal Reserve Bank (FRB). The descriptive statistics for the variables that were used are presented in Table 2.

**Table 2.** Descriptive statistics.

| | N | Mean | SD | Median | Trimmed | Mad | Min. | Max. | Range | Skew | Kurtosis | SE |
|---|---|---|---|---|---|---|---|---|---|---|---|---|
| Variable | 109 | 55 | 31.61 | 55 | 55 | 40.03 | 1 | 109 | 108 | 0 | −1.23 | 3.03 |
| Brazil PMI | 109 | 0 | 0.05 | 0 | 0 | 0.03 | −0.27 | 0.21 | 0.48 | −0.93 | 9.76 | 0 |
| Russia PMI | 109 | 0 | 0.05 | 0 | 0 | 0.03 | −0.4 | 0.2 | 0.6 | −3.65 | 31.9 | 0 |
| India PMI | 109 | 0 | 0.07 | 0 | 0 | 0.03 | −0.63 | 0.17 | 0.8 | −6.91 | 62.72 | 0.01 |
| China PMI | 109 | 0 | 0.04 | 0 | 0 | 0.01 | −0.33 | 0.25 | 0.58 | −2.66 | 44.85 | 0 |
| SA PMI | 109 | 0 | 0.05 | 0 | 0 | 0.04 | −0.19 | 0.15 | 0.34 | −0.43 | 2.68 | 0 |
| USA PMI | 109 | 0 | 0.04 | 0 | 0 | 0.02 | −0.26 | 0.15 | 0.42 | −2.55 | 23.48 | 0 |
| Brazil SPMI | 109 | 0 | 0.06 | 0 | 0 | 0.04 | −0.35 | 0.12 | 0.47 | −2.05 | 11.2 | 0.01 |
| Russia SPMI | 109 | −0.01 | 0.08 | 0 | 0 | 0.04 | −0.73 | 0.09 | 0.82 | −6.13 | 47.74 | 0.01 |
| India SPMI | 109 | −0.07 | 0.4 | −0.01 | −0.02 | 0.08 | −2.29 | 1.06 | 3.35 | −2.94 | 13.58 | 0.04 |
| China SPMI | 109 | 0 | 0.08 | 0 | 0 | 0.02 | −0.69 | 0.16 | 0.84 | −6.6 | 58.25 | 0.01 |
| SA SPMI | 109 | 0 | 0.06 | 0 | 0 | 0.06 | −0.22 | 0.18 | 0.4 | −0.5 | 1.98 | 0.01 |
| USA SPMI | 109 | 0 | 0.05 | 0 | 0 | 0.03 | −0.28 | 0.14 | 0.42 | −1.68 | 9.22 | 0 |
| Brazil UR | 109 | 0 | 0.05 | 0 | 0 | 0.05 | −0.11 | 0.1 | 0.21 | 0.15 | −0.72 | 0 |
| Russia UR | 109 | 0 | 0.05 | 0 | 0.01 | 0.04 | −0.34 | 0.1 | 0.45 | −2.75 | 16.43 | 0 |
| India UR | 109 | −0.01 | 0.21 | 0 | 0 | 0 | −1.7 | 1.21 | 2.91 | −3.09 | 47.76 | 0.02 |
| China UR | 109 | 0 | 0.04 | 0 | 0 | 0.01 | −0.1 | 0.24 | 0.33 | 2.58 | 12.99 | 0 |
| SA UR | 109 | 0 | 0.02 | 0 | 0 | 0.01 | −0.12 | 0.09 | 0.22 | −0.94 | 14.18 | 0 |
| USA UR | 109 | −0.01 | 0.08 | −0.01 | −0.01 | 0.03 | −0.23 | 0.76 | 0.99 | 6.79 | 62.76 | 0.01 |
| WTI | 109 | −0.01 | 0.05 | 0.01 | 0 | 0.03 | −0.21 | 0.06 | 0.27 | −1.74 | 4.16 | 0 |
| BDI | 109 | −0.01 | 0.09 | 0 | −0.01 | 0.03 | −0.45 | 0.25 | 0.7 | −1.28 | 7.22 | 0.01 |
| WEI | 109 | 0 | 0.05 | 0.01 | 0.01 | 0.04 | −0.18 | 0.15 | 0.32 | −0.69 | 1.1 | 0.01 |
| CEF | 109 | 0.01 | 0.15 | 0.02 | 0.02 | 0.09 | −0.52 | 0.35 | 0.88 | −1.2 | 2.8 | 0.01 |
| FBH | 109 | −0.01 | 0.08 | 0 | 0 | 0.01 | −0.74 | 0.12 | 0.86 | −6.94 | 59.49 | 0.01 |

### *4.2. Results*

Tables A1 and A2 in Appendix A present the conditional and partial correlations, respectively. Tables A4 and A5 provide the results of the ADF and KPSS stationary tests, respectively.

To assess the spillover and dynamic interconnected relationship between the six countries, Table A6 reports the findings of the static spillover connectedness between 2014 and 2023. The spillover contributions from (to) other factors vary between 43.11% (22.6%) and 95.88% (304.0%). Unemployment in South Africa has the lowest contribution from the other factors, while unemployment in India has the highest contribution from the others. The PMI in China is the highest transmitter of shocks, while the SPMI in India is the highest primary receiver of shocks.

The net pairwise directional connectedness (NPDC) plots in Figure 2 show the strengths and directions of the net spillover over time. The arrows pointing from one variable to another indicate positive net spillovers, while the arrow thickness defines the strength of the spillover. The color of the node defines the nature of the contributors and recipients.

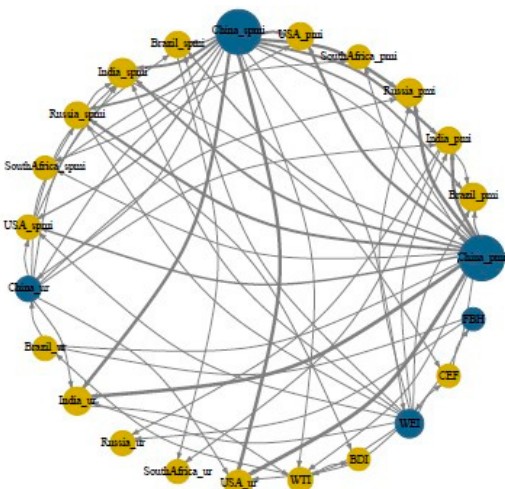

**Figure 2.** NPDC measure plot of the QVAR connectedness analysis.

Figure 2 shows that China's PMI and SPMI have the highest spillover to the other variables. The FBH, WEI, and unemployment in China are the other net transmitters. Figure A1 provides the NPDC plots across the quantiles. Clearly, the intensity of the spillover between the factors is much higher in the quantiles below 0.30 and above 0.85. Figure A2 in Appendix A presents the time series with abrupt changes between 2014 and 2023. Change points can be viewed as sudden fluctuations in time series representing transitions between states.

Figures A3–A7 in Appendix A present the wavelet coherence heatmaps. They allowed for the identification of significant, non-stationary stochastic patterns from these multivariate, nonlinear, time series. The degree of wavelet coherence (WC) is presented in the legend at the right of each heatmap and has values from 0.99 to 1.01 and periods between 4 and 32. Blue denotes a weak coherence, while red denotes a strong coherence. Figures A3–A7 show the WC maps via a contour plot. The vertical axis represents the frequencies as periods, which can be converted based on the windows to time units in days. The timeline in each map is shown on the horizontal axis from 2014 to 2023. One important finding that we may derive from the coherence maps displayed in Figures A3–A7 is the strong co-movement in the global supply chain at almost all the times and frequencies. It is interesting to note that the combined indexes had strong effects on the WTI, BDI, WEI, and FBH changes except for the CEF change. From these findings, one may infer a high time-frequency dependence and a causal effect between the factors and global supply chain. An exception, i.e., the low co-movement between the CEF and other variables, was found for the period 2014–2023. Another interesting finding is that the co-movement between the unemployment and other factors was weaker compared to the co-movement between the PMI and SPMI, particularly up to 2017.

### 4.3. Robustness Analysis

We examined the net directional accuracy of the factors in the system across a sequence of quantiles to analyze the sensitivity of the dynamic connectedness analysis. These results are reported in Tables 3 and 4. The total spillover contribution from (to) the other factors is almost identical across the sequence of quantiles, but the standard deviation (STDEV) in the spillover contribution to the other factors (Table 3) is greater than that in the spillover contribution from the other factors (Table 4).

These findings show a strong dynamic interconnection within the system. However, the standard deviation reaches its peak in the quantile at 0.50 (Figure 3). Figure 3 is constructed using the standard deviation of the total spillover contribution from (to) the other factors, as presented in Table 3 and 4, respectively. For the quantile values between 0.35 and 0.70, the total spillover contribution from the other factors is very stable, while the total spillover contribution to the other factors increases by 43.4% within the same range of quantiles.

**Table 3.** Simulation across a sequence of quantiles for spillover contributions to the other factors.

| FROM | 0.05 | 0.10 | 0.15 | 0.20 | 0.25 | 0.30 | 0.35 | 0.40 | 0.45 | 0.50 | 0.55 | 0.60 | 0.65 | 0.70 | 0.75 | 0.80 | 0.85 | 0.90 | 0.95 |
|---|---|---|---|---|---|---|---|---|---|---|---|---|---|---|---|---|---|---|---|
| China_pmi | 94.63 | 92.44 | 90.39 | 90.70 | 62.84 | 59.30 | 57.19 | 56.08 | 55.19 | 50.83 | 51.76 | 53.31 | 50.90 | 54.04 | 55.11 | 60.35 | 65.90 | 68.95 | 87.86 |
| Brazil_pmi | 96.74 | 95.18 | 87.07 | 86.54 | 82.34 | 79.97 | 83.50 | 81.34 | 82.53 | 83.39 | 82.35 | 83.74 | 84.43 | 86.43 | 82.90 | 83.30 | 86.42 | 88.33 | 95.04 |
| India_pmi | 90.66 | 95.24 | 93.52 | 93.84 | 89.67 | 86.89 | 88.41 | 89.72 | 90.33 | 90.79 | 89.06 | 93.65 | 91.68 | 94.03 | 91.93 | 89.52 | 90.02 | 90.74 | 94.15 |
| USA_pmi | 95.54 | 91.23 | 91.90 | 86.54 | 79.42 | 81.78 | 84.47 | 83.38 | 81.70 | 83.29 | 83.97 | 83.77 | 84.75 | 90.05 | 89.96 | 91.12 | 91.94 | 91.89 | 93.42 |
| China_spmi | 94.00 | 93.22 | 94.51 | 96.37 | 80.56 | 72.91 | 74.45 | 74.79 | 72.77 | 69.67 | 63.43 | 67.24 | 66.17 | 75.29 | 74.56 | 76.76 | 83.69 | 85.05 | 92.42 |
| Brazil_spmi | 95.44 | 95.50 | 89.66 | 91.26 | 81.64 | 78.05 | 74.01 | 70.53 | 74.00 | 76.89 | 74.65 | 75.13 | 79.43 | 84.58 | 82.16 | 83.96 | 89.18 | 90.51 | 91.75 |
| India_spmi | 96.64 | 95.86 | 93.61 | 94.69 | 92.15 | 86.40 | 87.30 | 84.13 | 84.54 | 85.38 | 85.47 | 86.60 | 86.24 | 87.40 | 88.20 | 87.71 | 91.03 | 94.56 | 96.71 |
| USA_spmi | 96.61 | 93.17 | 94.62 | 89.00 | 81.56 | 76.96 | 77.05 | 75.50 | 75.29 | 72.51 | 70.71 | 71.96 | 75.32 | 84.90 | 82.66 | 83.84 | 85.78 | 87.98 | 92.35 |
| China_uemp | 92.66 | 93.00 | 92.38 | 93.67 | 80.08 | 62.66 | 57.64 | 46.71 | 46.84 | 51.60 | 53.74 | 57.20 | 55.23 | 62.15 | 65.99 | 71.00 | 77.82 | 82.14 | 94.67 |
| Brazil_uemp | 77.78 | 91.40 | 89.34 | 91.63 | 75.58 | 56.14 | 40.31 | 39.43 | 40.85 | 64.61 | 61.25 | 54.26 | 64.67 | 80.65 | 81.22 | 84.16 | 90.67 | 89.65 | 95.80 |
| India_uemp | 92.60 | 92.55 | 91.76 | 90.61 | 87.44 | 82.18 | 77.59 | 64.82 | 50.99 | 50.46 | 49.10 | 48.85 | 49.20 | 53.54 | 59.12 | 63.89 | 76.35 | 85.67 | 96.35 |
| USA_uemp | 94.67 | 95.00 | 93.29 | 93.63 | 90.34 | 54.49 | 48.18 | 50.62 | 47.56 | 44.19 | 45.39 | 44.96 | 45.73 | 48.91 | 50.51 | 48.97 | 60.13 | 69.42 | 99.05 |
| WTI | 92.24 | 91.82 | 92.47 | 91.12 | 69.51 | 76.42 | 76.76 | 74.09 | 77.89 | 76.75 | 72.60 | 77.99 | 77.56 | 80.99 | 84.16 | 86.97 | 88.73 | 89.59 | 92.86 |
| BDI | 82.33 | 92.03 | 91.80 | 93.19 | 79.34 | 75.90 | 75.58 | 75.55 | 73.67 | 75.45 | 69.85 | 77.75 | 75.70 | 86.04 | 81.57 | 85.18 | 89.15 | 91.89 | 91.33 |
| WEI | 94.17 | 91.18 | 88.93 | 91.24 | 84.97 | 84.77 | 84.27 | 70.60 | 78.13 | 83.88 | 77.10 | 74.23 | 70.85 | 77.31 | 77.99 | 81.80 | 88.02 | 89.89 | 94.38 |
| CEF | 92.34 | 89.74 | 86.25 | 87.21 | 73.65 | 79.68 | 57.67 | 57.19 | 66.21 | 61.76 | 53.46 | 60.46 | 56.52 | 62.36 | 76.35 | 81.55 | 89.88 | 90.57 | 91.53 |
| STDEV | 5.21 | 1.84 | 2.52 | 2.90 | 7.73 | 10.66 | 14.70 | 14.59 | 15.42 | 14.60 | 14.10 | 14.70 | 14.59 | 14.51 | 12.45 | 11.80 | 9.39 | 7.44 | 2.65 |

**Table 4.** Simulation across a sequence of quantiles for spillover contributions from the other factors.

| TO | China_pmi | Brazil_pmi | India_pmi | USA_pmi | China_spmi | Brazil_spmi | India_spmi | USA_spmi | China_uemp | Brazil_uemp | India_uemp | USA_uemp | WTI | BDI | WEI | CEF | STDEV |
|---|---|---|---|---|---|---|---|---|---|---|---|---|---|---|---|---|---|
| 0.05 | 42.43 | 52.96 | 109.51 | 93.35 | 73.75 | 66.11 | 39.18 | 61.89 | 154.79 | 211.31 | 94.22 | 61.90 | 81.86 | 184.63 | 77.18 | 73.98 | 49.87 |
| 0.10 | 96.00 | 58.64 | 79.52 | 118.34 | 88.37 | 57.91 | 58.01 | 101.90 | 98.58 | 103.22 | 100.07 | 71.29 | 101.55 | 109.23 | 102.40 | 143.54 | 23.39 |
| 0.15 | 83.73 | 88.41 | 88.45 | 100.12 | 83.17 | 78.66 | 63.39 | 71.07 | 71.01 | 121.75 | 88.30 | 84.43 | 84.05 | 85.55 | 115.08 | 154.31 | 22.48 |
| 0.20 | 58.80 | 123.92 | 77.98 | 142.80 | 43.37 | 86.29 | 66.91 | 98.29 | 69.75 | 80.25 | 102.66 | 78.53 | 106.08 | 73.04 | 104.86 | 147.70 | 29.11 |
| 0.25 | 131.49 | 85.74 | 65.64 | 106.46 | 72.06 | 95.61 | 45.29 | 118.14 | 67.26 | 58.00 | 77.68 | 53.51 | 94.24 | 40.75 | 70.79 | 108.46 | 26.51 |
| 0.30 | 197.07 | 56.48 | 45.83 | 71.91 | 100.27 | 75.39 | 14.90 | 102.74 | 49.65 | 39.24 | 61.52 | 137.96 | 69.78 | 28.46 | 51.25 | 92.05 | 44.90 |
| 0.35 | 180.45 | 76.37 | 49.08 | 49.73 | 108.65 | 79.93 | 19.61 | 69.79 | 36.01 | 18.45 | 75.67 | 159.47 | 46.07 | 34.44 | 49.72 | 90.96 | 45.99 |
| 0.40 | 163.29 | 68.58 | 46.25 | 38.35 | 104.74 | 73.01 | 22.85 | 46.69 | 33.69 | 15.64 | 115.74 | 170.40 | 48.73 | 39.25 | 38.86 | 68.41 | 46.92 |
| 0.45 | 187.01 | 54.85 | 40.11 | 27.70 | 111.78 | 70.25 | 25.94 | 37.15 | 29.26 | 15.46 | 148.06 | 201.49 | 32.90 | 30.69 | 36.83 | 49.01 | 59.90 |
| 0.50 | 223.89 | 43.97 | 25.40 | 23.57 | 126.46 | 51.07 | 15.62 | 29.65 | 47.27 | 13.25 | 172.06 | 234.48 | 23.83 | 29.31 | 30.08 | 31.56 | 75.14 |
| 0.55 | 167.24 | 41.74 | 25.14 | 31.88 | 109.29 | 52.45 | 22.25 | 33.59 | 34.63 | 11.13 | 185.45 | 239.53 | 25.67 | 33.34 | 39.21 | 31.37 | 69.10 |
| 0.60 | 162.50 | 45.73 | 19.15 | 30.37 | 103.22 | 53.61 | 22.33 | 32.95 | 36.02 | 16.23 | 192.09 | 243.37 | 33.64 | 27.85 | 62.41 | 29.65 | 69.33 |
| 0.65 | 123.69 | 37.92 | 24.06 | 36.82 | 96.30 | 41.81 | 27.51 | 34.13 | 31.50 | 23.56 | 205.95 | 250.28 | 39.58 | 35.67 | 63.91 | 41.70 | 67.95 |
| 0.70 | 143.19 | 40.83 | 27.11 | 53.69 | 120.06 | 40.12 | 24.95 | 34.76 | 34.47 | 33.56 | 204.32 | 229.84 | 53.81 | 42.63 | 76.75 | 48.61 | 64.39 |
| 0.75 | 121.17 | 55.41 | 42.80 | 65.59 | 102.30 | 70.93 | 27.28 | 50.72 | 35.47 | 57.65 | 181.89 | 187.73 | 51.21 | 53.66 | 70.64 | 49.96 | 48.23 |
| 0.80 | 105.93 | 65.76 | 54.74 | 74.56 | 110.26 | 73.58 | 24.90 | 65.46 | 43.81 | 64.50 | 158.16 | 165.09 | 66.51 | 59.16 | 67.08 | 60.56 | 38.11 |
| 0.85 | 85.72 | 85.53 | 74.86 | 77.01 | 106.87 | 95.07 | 38.76 | 83.70 | 27.69 | 87.64 | 137.74 | 111.57 | 104.38 | 67.82 | 100.06 | 60.29 | 27.28 |
| 0.90 | 86.07 | 99.58 | 93.74 | 94.23 | 117.52 | 86.21 | 41.44 | 98.22 | 42.93 | 99.01 | 113.69 | 60.09 | 122.68 | 60.19 | 105.06 | 66.20 | 25.38 |
| 0.95 | 187.96 | 64.10 | 86.11 | 90.95 | 133.23 | 113.60 | 70.31 | 105.93 | 61.54 | 51.14 | 61.07 | 15.87 | 107.06 | 136.08 | 87.78 | 126.94 | 41.28 |

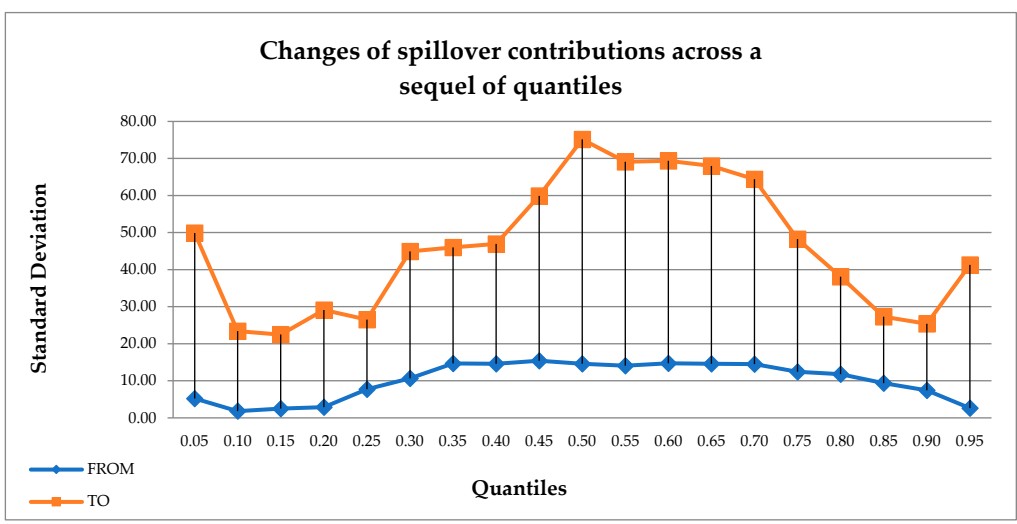

**Figure 3.** Changes in spillover contributions across a sequence of quantiles.

*4.4. Managerial Implications*

The empirical results on causality, cointegration, and dynamic connectedness provide several insights for managers and policymakers. First, policy decisions solely based on the correlation of two variables can be misleading owing to statistical noise consisting of residuals and errors in measurement and sampling. Second, it is important to pay close attention to the network structure and dynamic interconnection of spillovers among countries. The dynamics of the net pairwise directional connectedness of the spillover throughout countries, as shown by their arrows pointing from one factor to another, indicate positive net spillovers, while the arrow thickness defines the strength of the spillover. Moreover, the color of the factor node defines the nature of the contributors and recipients. The net pairwise directional connectedness figures show that the network of net pairwise directional spillovers among the BRICS countries differs across the quantiles. Our framework can be used by managers to make design decisions based on their supply chains and by policymakers at the time of designing economic policies in response to external shocks.

**5. Conclusions**

Following the challenges magnified by the COVID-19 pandemic, both public and private organizations faced capacity disruptions along their supply chains, resulting in significantly increased lead times and shortages due to lockdowns and logistics-related problems. Encountering the vulnerabilities of over-integrated supply-chain networks, business executives and policymakers began to reformulate their supply-chain designs and find ways to build resiliency into them.

The findings of this study enable business executives and policymakers to develop strategies for designing supply chains by considering the market dynamics under extenuating conditions. The results of the net pairwise directional spillovers between the investigated indicators present a complex pattern of interrelationships, particularly for extreme shocks. This study provides a conceptual framework to help understand the impact of large external shocks, such as the COVID-19 pandemic, on key production and supply-chain indicators. A conceptual framework that integrated correlation and dynamic connectedness analyses was implemented to examine the relationships between the PMI, SPMI, UR, BDI, WEI, CEF, FBH, and WTI for Brazil, Russia, India, China, South Africa, and the U.S. The results not only revealed the spillover and correlation relationships between these factors but also showed how these relations evolved following the impact of the worldwide pandemic.

One important finding that we may derive is the strong co-movement in the global supply chain at almost all the times. The results revealed a high time-frequency dependence

and a causal effect between the investigated factors and global supply chains. Another interesting finding is that the co-movement between the unemployment and other factors was weaker compared to the co-movement between the PMI and SPMI, particularly up to 2017.

There are inherent limitations on our study mainly owing to the availability of data across countries. A case in point is the purchasing managers' index (PMI), which shows the direction of economic trends in the service and manufacturing industries and can later be added for the remaining investigated countries.

One of the future directions of this research will be toward expanding the geographic scope of the countries involved to better assess the spillover effect on the global supply chains. Another direction will be conducting an industry-specific analysis of critical supply chains, such as food and medicine. The second direction will provide decisionmakers at the organizational level with strategies for redesigning their supply chains and choosing suppliers.

**Author Contributions:** Methodology, H.W.; Software, H.W.; Formal analysis, H.W.; Data curation, L.S.; Writing—original draft, L.S.; Writing—review & editing, J.O. All authors have read and agreed to the published version of the manuscript.

**Funding:** This research received no external funding.

**Institutional Review Board Statement:** Not applicable.

**Informed Consent Statement:** Not applicable.

**Data Availability Statement:** The data that support the findings of this study are available at https://www.investing.com.

**Conflicts of Interest:** We confirm that there are no known conflict of interest associated with this publication and that there has been no significant financial support for this work that could have influenced its outcome.



# Appendix A

**Table A1.** Conditional correlations.

| | Brazil pmi | Russia pmi | India pmi | China pmi | SA pmi | USA pmi | Brazil spmi | Russia spmi | India spmi | China spmi | SA spmi | USA spmi | Brazil ur | Russia ur | India ur | China ur | SA ur | USA ur | WTI | BDI | WEI | CEF | FBH |
|---|---|---|---|---|---|---|---|---|---|---|---|---|---|---|---|---|---|---|---|---|---|---|---|
| Brazil pmi | 1.000 | 0.074 | 0.331 | 0.015 | 0.019 | 0.237 | 0.429 | 0.362 | 0.033 | 0.095 | −0.010 | 0.180 | 0.077 | −0.009 | −0.306 | 0.020 | −0.061 | −0.331 | 0.054 | −0.077 | 0.015 | −0.097 | −0.152 |
| Russia pmi | 0.074 | 1.000 | 0.171 | −0.096 | −0.129 | 0.072 | 0.296 | 0.019 | 0.139 | −0.009 | −0.077 | 0.169 | 0.039 | −0.064 | −0.119 | 0.022 | −0.024 | −0.219 | 0.180 | −0.076 | −0.264 | 0.083 | 0.073 |
| India pmi | 0.331 | 0.171 | 1.000 | −0.069 | −0.010 | 0.224 | 0.146 | 0.493 | −0.137 | 0.053 | 0.148 | 0.350 | −0.125 | 0.123 | −0.472 | 0.026 | −0.292 | −0.520 | 0.009 | −0.162 | −0.056 | −0.078 | −0.109 |
| China pmi | 0.015 | −0.096 | −0.069 | 1.000 | −0.041 | −0.019 | −0.039 | 0.025 | −0.043 | 0.894 | 0.077 | 0.222 | −0.043 | 0.045 | −0.101 | −0.472 | 0.003 | 0.021 | 0.070 | −0.023 | 0.430 | 0.112 | 0.109 |
| SA pmi | 0.019 | −0.129 | −0.010 | −0.041 | 1.000 | 0.063 | 0.028 | 0.093 | −0.128 | 0.082 | 0.881 | −0.010 | −0.046 | −0.161 | 0.000 | 0.013 | 0.155 | −0.142 | −0.049 | 0.213 | −0.014 | 0.156 | −0.034 |
| USA pmi | 0.237 | 0.072 | 0.224 | −0.019 | 0.063 | 1.000 | 0.297 | 0.333 | −0.001 | 0.082 | 0.052 | 0.499 | 0.164 | −0.086 | −0.314 | 0.065 | 0.024 | −0.408 | 0.089 | −0.072 | 0.280 | −0.163 | −0.155 |
| Brazil spmi | 0.429 | 0.296 | 0.146 | −0.039 | 0.028 | 0.297 | 1.000 | 0.206 | 0.110 | 0.082 | −0.015 | 0.324 | 0.055 | −0.102 | −0.292 | 0.212 | −0.043 | −0.301 | 0.126 | −0.267 | 0.083 | −0.059 | −0.210 |
| Russia spmi | 0.362 | 0.019 | 0.493 | 0.025 | 0.093 | 0.333 | 0.206 | 1.000 | 0.036 | 0.169 | 0.157 | 0.433 | −0.320 | −0.079 | −0.381 | −0.114 | 0.098 | −0.452 | 0.136 | 0.071 | 0.106 | −0.085 | 0.088 |
| India spmi | 0.033 | 0.139 | −0.137 | −0.043 | −0.128 | −0.001 | 0.110 | 0.036 | 1.000 | −0.012 | −0.141 | 0.037 | 0.030 | −0.028 | 0.238 | −0.099 | 0.002 | 0.183 | 0.041 | −0.072 | −0.043 | −0.339 | −0.180 |
| China spmi | 0.095 | −0.009 | 0.053 | 0.894 | 0.082 | 0.082 | 0.082 | 0.169 | −0.012 | 1.000 | 0.186 | 0.257 | −0.063 | 0.009 | −0.146 | −0.474 | 0.055 | −0.080 | −0.025 | −0.047 | 0.374 | 0.069 | 0.027 |
| SA spmi | −0.010 | −0.077 | 0.148 | 0.077 | 0.881 | 0.052 | −0.015 | 0.157 | −0.141 | 0.186 | 1.000 | 0.098 | −0.256 | −0.119 | −0.153 | −0.077 | 0.100 | −0.254 | 0.012 | 0.159 | 0.018 | 0.208 | 0.041 |
| USA spmi | 0.180 | 0.169 | 0.350 | 0.222 | −0.010 | 0.499 | 0.324 | 0.433 | 0.037 | 0.257 | 0.098 | 1.000 | −0.112 | −0.157 | −0.318 | −0.054 | −0.085 | −0.411 | 0.206 | −0.081 | 0.198 | −0.027 | 0.000 |
| Brazil ur | 0.077 | 0.039 | −0.125 | −0.043 | −0.046 | 0.164 | 0.055 | −0.320 | 0.030 | −0.063 | −0.256 | −0.112 | 1.000 | −0.218 | 0.091 | 0.277 | −0.013 | 0.205 | −0.052 | −0.146 | −0.092 | −0.322 | −0.378 |
| Russia ur | −0.009 | −0.064 | 0.123 | 0.045 | −0.161 | −0.086 | −0.102 | −0.079 | −0.028 | 0.009 | −0.119 | −0.157 | −0.218 | 1.000 | 0.022 | −0.073 | −0.257 | −0.097 | −0.144 | −0.022 | 0.042 | −0.039 | −0.095 |
| India ur | −0.306 | −0.119 | −0.472 | −0.101 | 0.000 | −0.314 | −0.292 | −0.381 | 0.238 | −0.146 | −0.153 | −0.318 | 0.091 | 0.022 | 1.000 | −0.204 | −0.028 | 0.622 | −0.245 | 0.065 | −0.180 | −0.008 | 0.039 |
| China ur | 0.020 | 0.022 | 0.026 | −0.472 | 0.013 | 0.065 | 0.212 | −0.114 | −0.099 | −0.474 | −0.077 | −0.054 | 0.277 | −0.073 | −0.204 | 1.000 | −0.027 | −0.121 | 0.253 | −0.062 | −0.187 | −0.106 | −0.128 |
| SA ur | −0.061 | −0.024 | −0.292 | 0.003 | 0.155 | 0.024 | −0.043 | 0.098 | 0.002 | 0.055 | 0.100 | −0.085 | −0.013 | −0.257 | −0.028 | −0.027 | 1.000 | −0.047 | −0.170 | 0.156 | 0.084 | 0.020 | 0.066 |
| USA ur | −0.331 | −0.219 | −0.520 | 0.021 | −0.142 | −0.408 | −0.301 | −0.452 | 0.183 | −0.080 | −0.254 | −0.411 | 0.205 | −0.097 | 0.622 | −0.121 | −0.047 | 1.000 | −0.080 | −0.124 | −0.010 | −0.060 | 0.195 |
| WTI | 0.054 | 0.180 | 0.009 | 0.070 | −0.049 | 0.089 | 0.126 | 0.136 | 0.041 | −0.025 | 0.012 | 0.206 | −0.052 | −0.144 | −0.245 | 0.253 | −0.170 | −0.080 | 1.000 | 0.110 | 0.207 | 0.083 | 0.162 |
| BDI | −0.077 | −0.076 | −0.162 | −0.023 | 0.213 | −0.072 | −0.267 | 0.071 | −0.072 | −0.047 | 0.159 | −0.081 | −0.146 | −0.022 | 0.065 | −0.062 | 0.156 | −0.124 | 0.110 | 1.000 | 0.096 | 0.232 | 0.220 |
| WEI | 0.015 | −0.264 | −0.056 | 0.430 | −0.014 | 0.280 | 0.083 | 0.106 | −0.043 | 0.374 | 0.018 | 0.198 | −0.092 | 0.042 | −0.180 | −0.187 | 0.084 | −0.010 | 0.207 | 0.096 | 1.000 | 0.114 | 0.135 |
| CEF | −0.097 | 0.083 | −0.078 | 0.112 | 0.156 | −0.163 | −0.059 | −0.085 | −0.339 | 0.069 | 0.208 | −0.027 | −0.322 | −0.039 | −0.008 | −0.106 | 0.020 | −0.060 | 0.083 | 0.232 | 0.114 | 1.000 | 0.567 |
| FBH | −0.152 | 0.073 | −0.109 | 0.109 | −0.034 | −0.155 | −0.210 | 0.088 | −0.180 | 0.027 | 0.041 | 0.000 | −0.378 | −0.095 | 0.039 | −0.128 | 0.066 | 0.195 | 0.162 | 0.220 | 0.135 | 0.567 | 1.000 |

**Table A2.** Partial correlations.

| | Brazil pmi | Russia pmi | India pmi | China pmi | SA pmi | USA pmi | Brazil spmi | Russia spmi | India spmi | China spmi | SA spmi | USA spmi | Brazil ur | Russia ur | India ur | China ur | SA ur | USA ur | WTI | BDI | WEI | CEF | FBH |
|---|---|---|---|---|---|---|---|---|---|---|---|---|---|---|---|---|---|---|---|---|---|---|---|
| Brazil pmi | −1.000 | −0.131 | 0.152 | 0.045 | 0.075 | 0.024 | 0.358 | 0.160 | 0.066 | −0.028 | −0.095 | −0.128 | 0.147 | 0.025 | −0.071 | −0.135 | −0.017 | −0.093 | 0.050 | 0.034 | −0.099 | 0.016 | 0.032 |
| Russia pmi | −0.131 | −1.000 | 0.155 | −0.218 | −0.087 | −0.008 | 0.305 | −0.242 | 0.189 | 0.210 | 0.008 | 0.055 | 0.195 | 0.058 | 0.022 | −0.250 | 0.145 | −0.216 | 0.322 | −0.008 | −0.352 | 0.053 | 0.264 |
| India pmi | 0.152 | 0.155 | −1.000 | −0.210 | −0.217 | −0.176 | −0.206 | 0.377 | −0.131 | 0.122 | 0.242 | 0.147 | 0.164 | 0.080 | −0.161 | −0.001 | −0.412 | −0.199 | −0.175 | −0.192 | 0.112 | −0.030 | −0.016 |
| China pmi | 0.045 | −0.218 | −0.210 | −1.000 | −0.158 | −0.258 | −0.129 | −0.182 | −0.049 | 0.873 | 0.079 | 0.236 | 0.206 | 0.110 | −0.061 | −0.218 | −0.061 | −0.127 | 0.239 | −0.070 | 0.148 | −0.020 | 0.162 |
| SA pmi | 0.075 | −0.087 | −0.217 | −0.158 | −1.000 | −0.009 | 0.164 | 0.189 | −0.109 | 0.079 | 0.913 | −0.054 | 0.358 | 0.056 | 0.215 | 0.048 | 0.017 | 0.023 | −0.064 | 0.170 | 0.042 | 0.069 | −0.044 |
| USA pmi | 0.024 | −0.008 | −0.176 | −0.258 | −0.009 | −1.000 | −0.015 | 0.110 | −0.025 | 0.136 | 0.031 | 0.367 | 0.328 | 0.043 | −0.005 | −0.100 | −0.065 | −0.292 | −0.014 | −0.132 | 0.350 | −0.123 | 0.104 |
| Brazil spmi | 0.358 | 0.305 | −0.206 | −0.129 | 0.164 | −0.015 | −1.000 | 0.047 | 0.075 | 0.148 | −0.157 | 0.174 | −0.141 | −0.097 | −0.096 | 0.235 | −0.127 | −0.049 | −0.066 | −0.286 | 0.207 | 0.098 | −0.176 |
| Russia spmi | 0.160 | −0.242 | 0.377 | −0.182 | 0.189 | 0.110 | 0.047 | −1.000 | 0.106 | 0.212 | −0.212 | 0.155 | −0.265 | −0.096 | −0.056 | −0.165 | 0.228 | −0.124 | 0.222 | 0.074 | −0.105 | −0.232 | 0.247 |
| India spmi | 0.066 | 0.189 | −0.131 | −0.049 | −0.109 | −0.025 | 0.075 | 0.106 | −1.000 | 0.021 | 0.090 | 0.055 | −0.068 | −0.005 | 0.178 | −0.061 | −0.002 | 0.112 | 0.064 | 0.036 | 0.051 | −0.252 | −0.116 |
| China spmi | −0.028 | 0.210 | 0.122 | 0.873 | 0.079 | 0.136 | 0.148 | 0.212 | 0.021 | −1.000 | 0.028 | −0.117 | −0.064 | −0.051 | 0.003 | −0.016 | 0.059 | 0.104 | −0.227 | 0.019 | 0.008 | 0.027 | −0.147 |
| SA spmi | −0.095 | 0.008 | 0.242 | 0.079 | 0.913 | 0.031 | −0.157 | −0.212 | 0.090 | 0.028 | −1.000 | 0.039 | −0.400 | −0.124 | −0.197 | −0.047 | 0.011 | −0.060 | 0.093 | −0.117 | −0.083 | −0.008 | 0.011 |
| USA spmi | −0.128 | 0.055 | 0.147 | 0.236 | −0.054 | 0.367 | 0.174 | 0.155 | 0.055 | −0.117 | 0.039 | −1.000 | −0.099 | −0.230 | 0.108 | 0.041 | −0.100 | −0.131 | 0.036 | −0.022 | −0.013 | −0.022 | 0.022 |
| Brazil ur | 0.147 | 0.195 | 0.164 | 0.206 | 0.358 | 0.328 | −0.141 | −0.265 | −0.068 | −0.064 | −0.400 | −0.099 | −1.000 | −0.314 | −0.062 | 0.284 | 0.003 | 0.239 | −0.073 | 0.038 | −0.071 | −0.104 | −0.289 |
| Russia ur | 0.025 | 0.058 | 0.080 | 0.110 | 0.056 | 0.043 | −0.097 | −0.096 | −0.005 | −0.051 | −0.124 | −0.230 | −0.314 | −1.000 | 0.071 | 0.085 | −0.257 | −0.156 | −0.164 | −0.013 | 0.127 | −0.082 | −0.087 |
| India ur | −0.071 | 0.022 | −0.161 | −0.061 | 0.215 | −0.005 | −0.096 | −0.056 | 0.178 | 0.003 | −0.197 | 0.108 | −0.062 | 0.071 | −1.000 | −0.181 | −0.101 | 0.382 | −0.144 | 0.104 | −0.148 | 0.098 | −0.096 |
| China ur | −0.135 | −0.250 | −0.001 | −0.218 | 0.048 | −0.100 | 0.235 | −0.165 | −0.061 | −0.016 | −0.047 | 0.041 | 0.284 | 0.085 | −0.181 | −1.000 | 0.073 | −0.147 | 0.354 | −0.061 | −0.120 | −0.076 | 0.140 |
| SA ur | −0.017 | 0.145 | −0.412 | −0.061 | 0.017 | −0.065 | −0.127 | 0.228 | −0.002 | 0.059 | 0.011 | −0.100 | 0.003 | −0.257 | −0.101 | 0.073 | −1.000 | −0.157 | −0.304 | −0.003 | 0.184 | −0.067 | 0.036 |
| USA ur | −0.093 | −0.216 | −0.199 | −0.127 | 0.023 | −0.292 | −0.049 | −0.124 | 0.112 | 0.104 | −0.060 | −0.131 | 0.239 | −0.156 | 0.382 | −0.147 | −0.157 | −1.000 | 0.103 | −0.334 | 0.160 | −0.238 | 0.424 |
| WTI | 0.050 | 0.322 | −0.175 | 0.239 | −0.064 | −0.014 | −0.066 | 0.222 | 0.064 | −0.227 | 0.093 | 0.036 | −0.073 | −0.164 | −0.144 | 0.354 | −0.304 | 0.103 | −1.000 | 0.128 | 0.245 | 0.018 | −0.036 |
| BDI | 0.034 | −0.008 | −0.192 | −0.070 | 0.170 | −0.132 | −0.286 | 0.074 | 0.036 | 0.019 | −0.117 | −0.022 | 0.038 | −0.013 | 0.104 | −0.061 | −0.003 | −0.334 | 0.128 | −1.000 | 0.148 | 0.044 | 0.136 |
| WEI | −0.099 | −0.352 | 0.112 | 0.148 | 0.042 | 0.350 | 0.207 | −0.105 | 0.051 | 0.008 | −0.083 | −0.013 | −0.071 | 0.127 | −0.148 | −0.120 | 0.184 | 0.160 | 0.245 | 0.148 | −1.000 | 0.099 | 0.043 |
| CEF | 0.016 | 0.053 | −0.030 | −0.020 | 0.069 | −0.123 | 0.098 | −0.232 | −0.252 | 0.027 | −0.008 | −0.022 | −0.104 | −0.082 | 0.098 | −0.076 | −0.067 | −0.238 | 0.018 | 0.044 | 0.099 | −1.000 | 0.496 |
| FBH | 0.032 | 0.264 | −0.016 | 0.162 | −0.044 | 0.104 | −0.176 | 0.247 | −0.116 | −0.147 | 0.011 | 0.022 | −0.289 | −0.087 | −0.096 | 0.140 | 0.036 | 0.424 | −0.036 | 0.136 | 0.043 | 0.496 | −1.000 |

**Table A3.** Mutual information.

| . | Brazil pmi | Russia pmi | India pmi | China pmi | SA pmi | USA pmi | Brazil spmi | Russia spmi | India spmi | China spmi | SA spmi | USA spmi | Brazil ur | Russia ur | India ur | China ur | SA ur | USA ur | WTI | BDI | WEI | CEF | FBH |
|---|---|---|---|---|---|---|---|---|---|---|---|---|---|---|---|---|---|---|---|---|---|---|---|
| Brazil pmi | | 0.010 | 0.000 | 0.033 | 0.038 | 0.001 | 0.000 | 0.000 | 0.001 | 0.000 | 0.060 | 0.032 | 0.000 | 0.030 | 0.027 | 0.000 | 0.056 | 0.000 | 0.012 | 0.012 | 0.049 | 0.000 | 0.029 |
| Russia pmi | 0.010 | | 0.000 | 0.000 | 0.000 | 0.000 | 0.023 | 0.111 | 0.070 | 0.000 | 0.000 | 0.051 | 0.000 | 0.000 | 0.086 | 0.000 | 0.000 | 0.055 | 0.000 | 0.063 | 0.000 | 0.000 | 0.070 |
| India pmi | 0.000 | 0.000 | | 0.073 | 0.000 | 0.000 | 0.019 | 0.000 | 0.029 | 0.016 | 0.006 | 0.073 | 0.000 | 0.057 | 0.000 | 0.017 | 0.000 | 0.000 | 0.000 | 0.000 | 0.023 | 0.016 | 0.025 |
| China pmi | 0.033 | 0.000 | 0.068 | | 0.007 | 0.000 | 0.090 | 0.067 | 0.039 | 0.062 | 0.049 | 0.000 | 0.000 | 0.000 | 0.131 | 0.000 | 0.000 | 0.078 | 0.026 | 0.000 | 0.067 | 0.000 | 0.000 |
| SA pmi | 0.037 | 0.000 | 0.000 | 0.004 | | 0.000 | 0.095 | 0.076 | 0.044 | 0.000 | 0.875 | 0.000 | 0.000 | 0.017 | 0.008 | 0.000 | 0.000 | 0.094 | 0.009 | 0.003 | 0.035 | 0.036 | 0.000 |
| USA pmi | 0.001 | 0.000 | 0.000 | 0.000 | 0.000 | | 0.053 | 0.107 | 0.065 | 0.000 | 0.000 | 0.037 | 0.000 | 0.000 | 0.098 | 0.036 | 0.026 | 0.012 | 0.008 | 0.035 | 0.011 | 0.061 | 0.067 |
| Brazil spmi | 0.000 | 0.023 | 0.018 | 0.092 | 0.095 | 0.054 | | 0.073 | 0.000 | 0.029 | 0.082 | 0.026 | 0.067 | 0.019 | 0.039 | 0.047 | 0.082 | 0.014 | 0.000 | 0.005 | 0.000 | 0.018 | 0.000 |
| Russia spmi | 0.000 | 0.111 | 0.000 | 0.064 | 0.076 | 0.107 | 0.072 | | 0.000 | 0.000 | 0.061 | 0.051 | 0.000 | 0.000 | 0.121 | 0.018 | 0.000 | 0.000 | 0.034 | 0.040 | 0.075 | 0.041 | 0.000 |
| India spmi | 0.001 | 0.070 | 0.027 | 0.042 | 0.044 | 0.065 | 0.000 | 0.000 | | 0.000 | 0.111 | 0.024 | 0.000 | 0.023 | 0.065 | 0.016 | 0.007 | 0.236 | 0.000 | 0.078 | 0.000 | 0.000 | 0.000 |
| China spmi | 0.000 | 0.000 | 0.018 | 0.060 | 0.000 | 0.000 | 0.029 | 0.000 | 0.000 | | 0.068 | 0.000 | 0.000 | 0.021 | 0.000 | 0.115 | 0.000 | 0.000 | 0.162 | 0.000 | 0.024 | 0.000 | 0.000 |
| SA spmi | 0.059 | 0.000 | 0.003 | 0.049 | 0.875 | 0.000 | 0.082 | 0.061 | 0.111 | 0.068 | | 0.000 | 0.038 | 0.032 | 0.000 | 0.000 | 0.000 | 0.046 | 0.038 | 0.003 | 0.030 | 0.002 | 0.000 |
| USA spmi | 0.031 | 0.050 | 0.073 | 0.000 | 0.000 | 0.037 | 0.026 | 0.052 | 0.024 | 0.000 | 0.000 | | 0.080 | 0.019 | 0.066 | 0.000 | 0.047 | 0.095 | 0.127 | 0.087 | 0.012 | 0.000 | 0.026 |
| Brazil ur | 0.000 | 0.000 | 0.000 | 0.000 | 0.000 | 0.000 | 0.070 | 0.000 | 0.000 | 0.000 | 0.046 | 0.081 | | 0.082 | 0.139 | 0.024 | 0.049 | 0.098 | 0.000 | 0.025 | 0.023 | 0.000 | 0.058 |
| Russia ur | 0.028 | 0.000 | 0.056 | 0.000 | 0.020 | 0.000 | 0.021 | 0.000 | 0.023 | 0.019 | 0.034 | 0.021 | 0.084 | | 0.119 | 0.145 | 0.000 | 0.076 | 0.000 | 0.046 | 0.000 | 0.000 | 0.109 |
| India ur | 0.026 | 0.091 | 0.004 | 0.104 | 0.000 | 0.093 | 0.069 | 0.111 | 0.054 | 0.008 | 0.003 | 0.071 | 0.162 | 0.167 | | 0.108 | 0.083 | 0.266 | 0.003 | 0.197 | 0.174 | 0.021 | 0.088 |
| China ur | 0.021 | 0.000 | 0.019 | 0.000 | 0.000 | 0.000 | 0.030 | 0.020 | 0.000 | 0.128 | 0.000 | 0.000 | 0.049 | 0.117 | 0.129 | | 0.000 | 0.057 | 0.000 | 0.080 | 0.018 | 0.010 | 0.038 |
| SA ur | 0.055 | 0.000 | 0.000 | 0.000 | 0.000 | 0.028 | 0.083 | 0.000 | 0.001 | 0.000 | 0.000 | 0.047 | 0.047 | 0.000 | 0.126 | 0.000 | | 0.055 | 0.056 | 0.169 | 0.000 | 0.055 | 0.000 |
| USA ur | 0.000 | 0.057 | 0.000 | 0.070 | 0.094 | 0.015 | 0.017 | 0.000 | 0.234 | 0.000 | 0.047 | 0.086 | 0.085 | 0.075 | 0.211 | 0.000 | 0.050 | | 0.000 | 0.000 | 0.077 | 0.019 | 0.000 |
| WTI | 0.011 | 0.000 | 0.000 | 0.021 | 0.009 | 0.007 | 0.000 | 0.034 | 0.000 | 0.162 | 0.038 | 0.127 | 0.000 | 0.000 | 0.000 | 0.000 | 0.056 | 0.000 | | 0.000 | 0.000 | 0.000 | 0.037 |
| BDI | 0.012 | 0.064 | 0.000 | 0.000 | 0.003 | 0.035 | 0.005 | 0.040 | 0.078 | 0.000 | 0.003 | 0.087 | 0.023 | 0.048 | 0.205 | 0.093 | 0.164 | 0.000 | 0.000 | | 0.079 | 0.040 | 0.028 |
| WEI | 0.049 | 0.000 | 0.023 | 0.068 | 0.036 | 0.011 | 0.000 | 0.075 | 0.000 | 0.024 | 0.030 | 0.012 | 0.026 | 0.000 | 0.184 | 0.007 | 0.000 | 0.073 | 0.000 | 0.079 | | 0.042 | 0.000 |
| CEF | 0.000 | 0.000 | 0.016 | 0.001 | 0.036 | 0.061 | 0.018 | 0.041 | 0.000 | 0.000 | 0.002 | 0.000 | 0.000 | 0.000 | 0.032 | 0.025 | 0.059 | 0.022 | 0.000 | 0.040 | 0.042 | | 0.029 |
| FBH | 0.029 | 0.070 | 0.025 | 0.000 | 0.000 | 0.067 | 0.000 | 0.000 | 0.000 | 0.000 | 0.000 | 0.026 | 0.056 | 0.103 | 0.107 | 0.057 | 0.000 | 0.000 | 0.037 | 0.028 | 0.000 | 0.029 | |

**Table A4.** Stationary test (ADF).

| Ticker | Brazil pmi | Russia pmi | India pmi | China pmi | SA pmi | USA pmi | Brazil spmi | Russia spmi | India spmi | China spmi | SA spmi | USA spmi | Brazil ur | Russia ur | India ur | China ur | SA ur | USA ur | WTI | BDI | WEI | CEF | FBH |
|---|---|---|---|---|---|---|---|---|---|---|---|---|---|---|---|---|---|---|---|---|---|---|---|
| ADF Statistic | −8.0469 | −8.3153 | −8.8978 | −9.13885 | −6.6905 | −8.47313 | −6.9228 | −6.8929 | −4.9355 | −7.2013 | −7.3135 | −8.30878 | −2.33445 | −7.4239 | −10.7696 | −6.8692 | −5.08827 | −8.13729 | −6.14428 | −4.54037 | −11.7518 | −4.1087 | −9.9894 |
| *p*-value | $1.7 \times 10^{-12}$ | $3.6 \times 10^{-13}$ | $1.2 \times 10^{-14}$ | $2.89 \times 10^{-15}$ | $4.1 \times 10^{-9}$ | $1.46 \times 10^{-13}$ | $1.1 \times 10^{-9}$ | $1.3 \times 10^{-9}$ | $2.9 \times 10^{-5}$ | $2.3 \times 10^{-10}$ | $1.2 \times 10^{-10}$ | $3.84 \times 10^{-13}$ | 0.161097706 | $6.6 \times 10^{-11}$ | $2.4 \times 10^{-19}$ | $1.5 \times 10^{-9}$ | $1.48 \times 10^{-5}$ | $1.05 \times 10^{-12}$ | $7.8 \times 10^{-8}$ | 0.00016612 | $1.2 \times 10^{-21}$ | 0.0009369 | $2.0 \times 10^{-17}$ |
| Lags Used | 1 | 3 | 1 | 2 | 5 | 1 | 3 | 0 | 2 | 3 | 5 | 1 | 12 | 0 | 1 | 3 | 5 | 1 | 1 | 10 | 0 | 4 | 0 |

**Table A5.** Stationary test (KPSS).

| Ticker | Brazil pmi | Russia pmi | India pmi | China pmi | SA pmi | USA pmi | Brazil spmi | Russia spmi | India spmi | China spmi | SA spmi | USA spmi | Brazil ur | Russia ur | India ur | China ur | SA ur | USA ur | WTI | BDI | WEI | CEF | FBH |
|---|---|---|---|---|---|---|---|---|---|---|---|---|---|---|---|---|---|---|---|---|---|---|---|
| KPSS Statistic | 0.03738 | 0.11609 | 0.0631 | 0.1330 | 0.0538 | 0.0488 | 0.028 | 0.134 | 0.1919 | 0.07382 | 0.0494 | 0.0363 | 0.4642 | 0.0997 | 0.1153 | 0.04799 | 0.04787 | 0.0335 | 0.2059 | 0.16330 | 0.0579 | 0.49591 | 0.16662 |
| *p*-value | $1.78 \times 10^{-12}$ | $3.69 \times 10^{-13}$ | $1.2 \times 10^{-14}$ | $2.9 \times 10^{-15}$ | $4.1 \times 10^{-9}$ | $1.46 \times 10^{-13}$ | $1.1 \times 10^{-9}$ | $1.3 \times 10^{-9}$ | $2.97 \times 10^{-5}$ | $2.36 \times 10^{-10}$ | $1.25 \times 10^{-10}$ | $3.8 \times 10^{-13}$ | 0.1610977 | $6.62 \times 10^{-11}$ | $2.40 \times 10^{-19}$ | $1.53 \times 10^{-9}$ | $1.48 \times 10^{-5}$ | $1.05 \times 10^{-12}$ | $7.8 \times 10^{-8}$ | 0.0001661 | $1.20 \times 10^{-21}$ | 0.000936945 | $2.01 \times 10^{-17}$ |
| Lags Used | 1 | 3 | 1 | 2 | 5 | 1 | 3 | 0 | 2 | 3 | 5 | 1 | 12 | 0 | 1 | 3 | 5 | 1 | 1 | 10 | 0 | 4 | 0 |

**Table A6.** Dynamic connectedness analysis.

| | Brazil pmi | Russia pmi | India pmi | China pmi | SA pmi | USA pmi | Brazil spmi | Russia spmi | India spmi | China spmi | SA spmi | USA spmi | Brazil ur | Russia ur | India ur | China ur | SA ur | USA ur | WTI | BDI | WEI | CEF | FBH | FROM |
|---|---|---|---|---|---|---|---|---|---|---|---|---|---|---|---|---|---|---|---|---|---|---|---|---|
| Brazil pmi | 18.6 | 1.77 | 5.16 | 15.3 | 0.62 | 1.61 | 6.46 | 4.14 | 0.52 | 17.7 | 1.33 | 3.21 | 1.48 | 1.11 | 2.46 | 3.3 | 1.19 | 3.01 | 0.5 | 2.47 | 4.49 | 1.87 | 1.82 | 81.45 |
| Russia pmi | 1.01 | 10.4 | 1.97 | 26.4 | 1.05 | 0.89 | 1.46 | 1.9 | 1.17 | 24.7 | 0.76 | 3.94 | 0.91 | 0.19 | 1.52 | 4.71 | 0.68 | 0.77 | 1.08 | 1.99 | 10.1 | 1.16 | 1.3 | 89.61 |
| India pmi | 1.33 | 1.5 | 7.75 | 26.3 | 0.43 | 0.7 | 1.5 | 2.66 | 0.8 | 25 | 0.82 | 3.21 | 1.38 | 0.55 | 2.29 | 6.53 | 1.45 | 2.2 | 0.67 | 1.78 | 7.65 | 2.11 | 1.37 | 92.25 |
| China pmi | 0.24 | 0.95 | 1.27 | 37.9 | 0.39 | 0.42 | 0.19 | 0.53 | 0.33 | 31.9 | 0.57 | 1.92 | 0.86 | 0.34 | 0.86 | 8.74 | 0.82 | 0.15 | 0.6 | 1.6 | 5.8 | 2.19 | 1.46 | 62.11 |
| SA pmi | 0.84 | 3.64 | 1.75 | 3.66 | 38 | 0.98 | 0.48 | 0.96 | 1.22 | 2.88 | 29 | 1.92 | 0.68 | 1.02 | 0.65 | 1.89 | 1.72 | 1.3 | 0.54 | 2.16 | 2.83 | 1.37 | 0.58 | 62.03 |
| USA pmi | 1.14 | 1.33 | 1.69 | 23.2 | 0.94 | 10.6 | 1.4 | 2.05 | 0.5 | 23.4 | 1.35 | 5.25 | 1.64 | 0.41 | 1.61 | 6.45 | 0.82 | 1.98 | 0.95 | 1.63 | 7.55 | 2.61 | 1.52 | 89.43 |
| Brazil spmi | 4.33 | 2.76 | 2.06 | 14.4 | 0.8 | 2.77 | 19.2 | 2.26 | 0.38 | 15.3 | 2.09 | 4.87 | 2.02 | 1.1 | 2.73 | 3.7 | 0.49 | 2.35 | 0.55 | 2.26 | 6.22 | 2.44 | 4.98 | 80.84 |
| Russia spmi | 2.31 | 0.95 | 2.96 | 22.7 | 0.49 | 1.98 | 2.5 | 9.94 | 0.39 | 22.9 | 1.02 | 4.79 | 2.25 | 0.53 | 3.05 | 4.45 | 0.97 | 3.52 | 1.08 | 1 | 7.59 | 1.76 | 0.82 | 90.06 |
| India spmi | 1.94 | 1.16 | 2.4 | 19.9 | 0.45 | 1.11 | 2.07 | 6.37 | 9.57 | 23 | 0.59 | 6.04 | 1.72 | 0.5 | 2.49 | 6.41 | 0.23 | 3.29 | 0.81 | 1.23 | 6.47 | 1.72 | 0.51 | 90.43 |
| China spmi | 0.82 | 0.81 | 2.42 | 28 | 0.83 | 0.99 | 0.71 | 2.15 | 0.25 | 35.5 | 1.76 | 3.12 | 1.11 | 0.27 | 1.2 | 8.13 | 1 | 0.44 | 0.4 | 2.05 | 5.49 | 1.72 | 0.86 | 64.48 |
| SA spmi | 0.83 | 3.11 | 1.8 | 6.49 | 24.7 | 0.73 | 0.96 | 1.43 | 1.07 | 6.11 | 32.2 | 1.31 | 4.1 | 0.46 | 1.22 | 2.46 | 1.3 | 2.24 | 0.63 | 1.68 | 2.36 | 1.92 | 0.86 | 67.78 |
| USA spmi | 1.68 | 1.8 | 2.36 | 18.8 | 0.68 | 4.22 | 2.02 | 3.24 | 1.13 | 18 | 1.46 | 15.9 | 1.52 | 0.53 | 2.25 | 5.06 | 0.53 | 2.97 | 2.26 | 1.09 | 7.68 | 2.3 | 2.52 | 84.11 |
| Brazil ur | 0.97 | 0.98 | 1.15 | 4.45 | 0.36 | 2.23 | 0.75 | 4.31 | 0.66 | 3.52 | 1.83 | 3.05 | 28.8 | 4.87 | 1.28 | 2.65 | 1.55 | 1.78 | 10.3 | 6.71 | 4.05 | 4.17 | 9.63 | 71.19 |

**Table A6.** *Cont.*

| | Brazil pmi | Russia pmi | India pmi | China pmi | SA pmi | USA pmi | Brazil spmi | Russia spmi | India spmi | China spmi | SA spmi | USA spmi | Brazil ur | Russia ur | India ur | China ur | SA ur | USA ur | WTI | BDI | WEI | CEF | FBH | FROM |
|---|---|---|---|---|---|---|---|---|---|---|---|---|---|---|---|---|---|---|---|---|---|---|---|---|
| Russia ur | 0.85 | 0.72 | 4.68 | 1.72 | 1.93 | 1.74 | 2.45 | 1.27 | 0.69 | 1.43 | 2.06 | 1.79 | 2.58 | 50.7 | 0.68 | 1.23 | 6.93 | 1.26 | 3.65 | 2.88 | 2.55 | 1.84 | 4.36 | 49.32 |
| India ur | 1.14 | 1.97 | 3.98 | 27.6 | 0.69 | 0.89 | 1.47 | 2.06 | 0.83 | 24.7 | 1.04 | 2.82 | 1.49 | 0.35 | 4.12 | 5.22 | 1.12 | 3.11 | 0.72 | 2.31 | 8.88 | 2.07 | 1.41 | 95.88 |
| China ur | 0.62 | 0.6 | 2.1 | 11.1 | 1.05 | 2.03 | 2.8 | 1.56 | 1.16 | 12.1 | 1.49 | 0.61 | 5.97 | 2.6 | 2.21 | 39.8 | 1.23 | 1.15 | 2.63 | 1.57 | 2.46 | 1.84 | 1.28 | 60.17 |
| SA ur | 2.16 | 1.17 | 4.84 | 0.46 | 2.54 | 0.92 | 3.68 | 0.94 | 0.97 | 0.71 | 1.97 | 1.41 | 2.62 | 6.67 | 0.19 | 0.42 | 56.9 | 0.45 | 2.07 | 1.94 | 1.18 | 2.49 | 3.29 | 43.11 |
| USA ur | 1.16 | 1.2 | 3.17 | 27.5 | 0.6 | 1.12 | 1.48 | 2.24 | 0.86 | 26 | 1.38 | 3.39 | 1.61 | 0.69 | 3.35 | 4.88 | 0.87 | 5.14 | 0.56 | 1.6 | 8.05 | 1.92 | 1.27 | 94.86 |
| WTI | 0.53 | 3.15 | 1.28 | 11 | 0.27 | 2.15 | 1.42 | 2.17 | 0.97 | 8.81 | 0.72 | 3.39 | 1.96 | 0.97 | 3.78 | 3.08 | 1.53 | 1 | 33.2 | 2.01 | 12.6 | 1.21 | 2.84 | 66.8 |
| BDI | 0.71 | 1.13 | 2.44 | 1.05 | 2.59 | 2.22 | 3.89 | 5.26 | 0.49 | 0.97 | 2.12 | 1.39 | 4.17 | 1.9 | 0.94 | 1.3 | 4.05 | 1.86 | 5.4 | 48.2 | 1.86 | 2.92 | 3.13 | 51.77 |
| WEI | 0.73 | 3.76 | 0.48 | 11.1 | 3.79 | 5.21 | 1.13 | 1.39 | 0.59 | 9.62 | 2.5 | 3.13 | 2.33 | 0.43 | 2.23 | 2.69 | 1.3 | 1.2 | 4.5 | 1.5 | 37.7 | 0.81 | 1.88 | 62.26 |
| CEF | 1.35 | 1.35 | 2.35 | 2.47 | 3.1 | 3.62 | 1.35 | 1.45 | 5.62 | 2.16 | 2.16 | 3.86 | 3.68 | 0.4 | 1.5 | 3 | 0.65 | 0.64 | 1.16 | 5.33 | 4.95 | 36.6 | 11.2 | 63.39 |
| FBH | 1.31 | 0.73 | 1.71 | 1.02 | 0.74 | 4.23 | 3.31 | 1.05 | 1.96 | 0.44 | 1.31 | 1.21 | 6.41 | 1.13 | 0.46 | 1.91 | 0.98 | 2.3 | 2.26 | 3.4 | 2.97 | 14.8 | 44.4 | 55.62 |
| TO | 28 | 36.6 | 54 | 304 | 49.1 | 42.8 | 43.5 | 51.4 | 22.6 | 301 | 59.3 | 65.7 | 52.5 | 27 | 39 | 88.2 | 31.4 | 39 | 43.3 | 50.2 | 124 | 57.2 | 58.9 | 1668.95 |
| Inc. Own | 46.6 | 47 | 61.8 | 342 | 87 | 53.3 | 62.6 | 61.3 | 32.1 | 337 | 91.5 | 81.6 | 81.3 | 77.7 | 43.1 | 128 | 88.3 | 44.1 | 76.5 | 98.4 | 161 | 93.8 | 103 | cTCI/TCI |
| NET | −53.4 | −53.1 | −38.2 | 242 | −13 | −46.7 | −37.4 | −38.7 | −67.9 | 237 | −8.49 | −18.5 | −18.7 | −22.3 | −56.9 | 28.1 | −11.7 | −55.9 | −23.5 | −1.58 | 61.4 | −6.17 | 3.29 | 75.86/72.56 |
| NPT | 4 | 12 | 16 | 18 | 8 | 11 | 7 | 11 | 8 | 19 | 12 | 16 | 9 | 3 | 9 | 16 | 9 | 12 | 5 | 11 | 16 | 10 | 11 | |

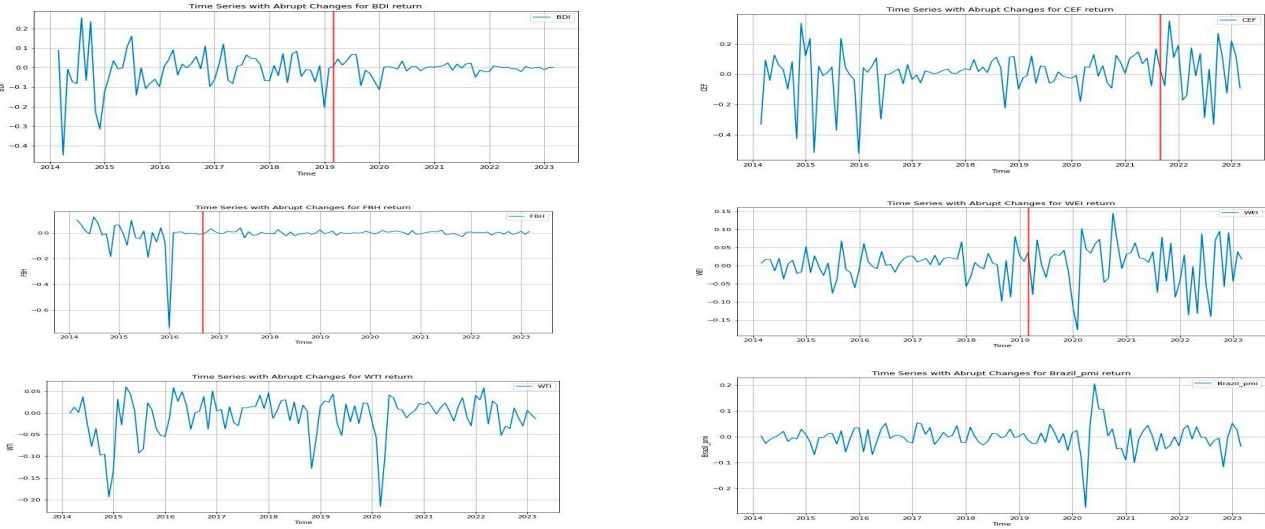

**Figure A1.** NPDC-measured plots of the QVAR connectedness analysis across quantiles.

**Figure A2.** *Cont.*

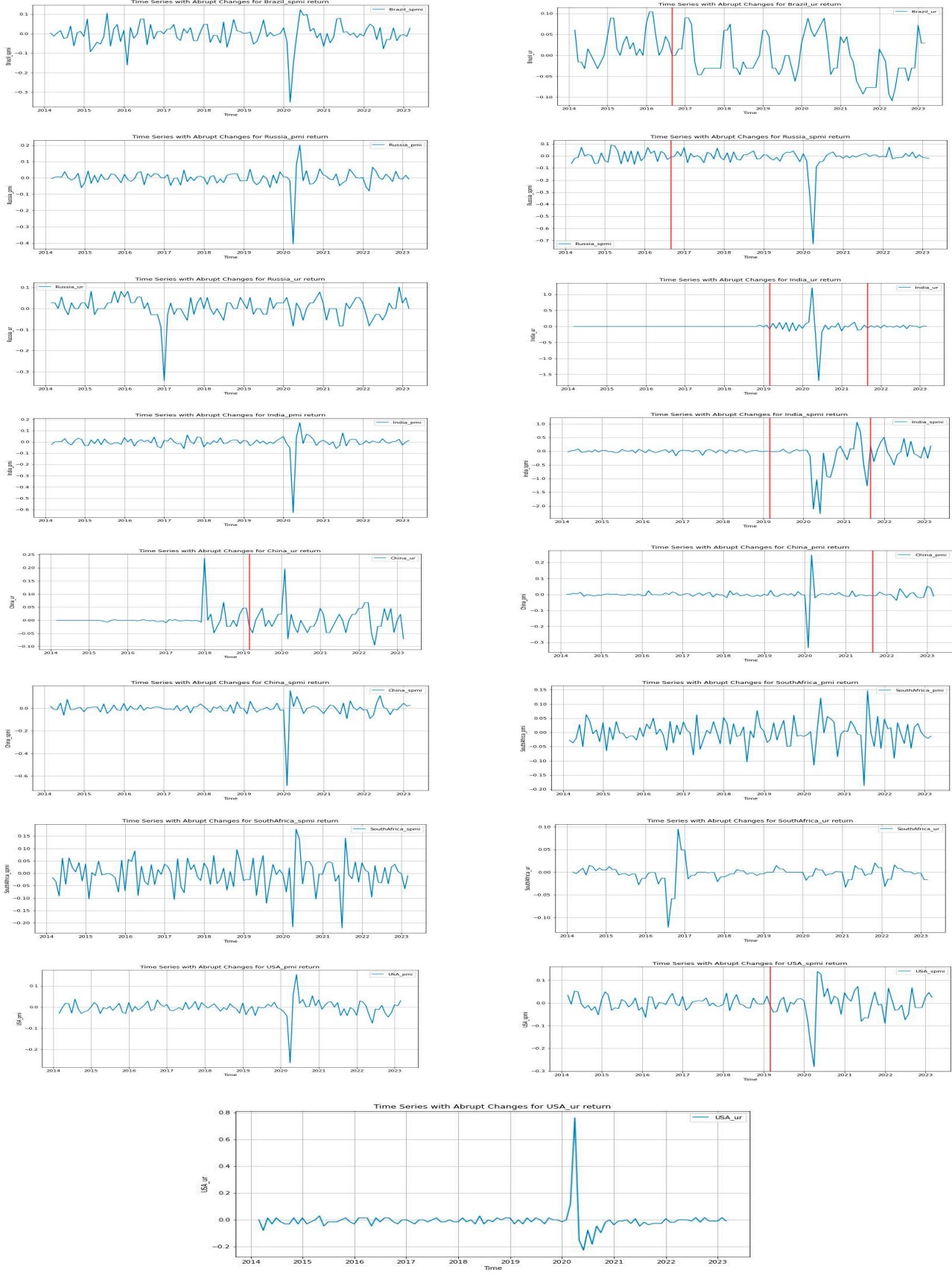

**Figure A2.** Abrupt changes in time series data.

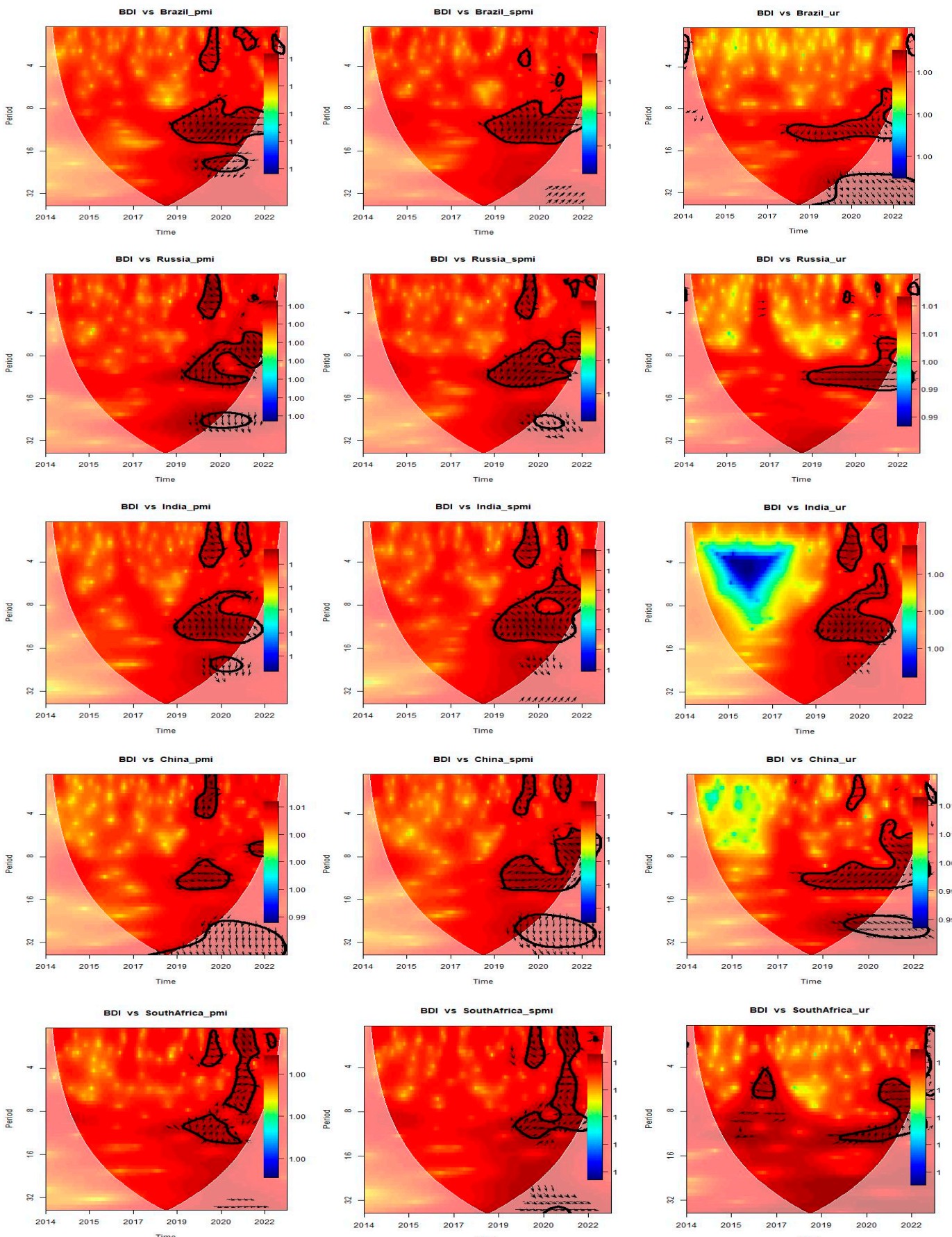

**Figure A3.** *Cont.*

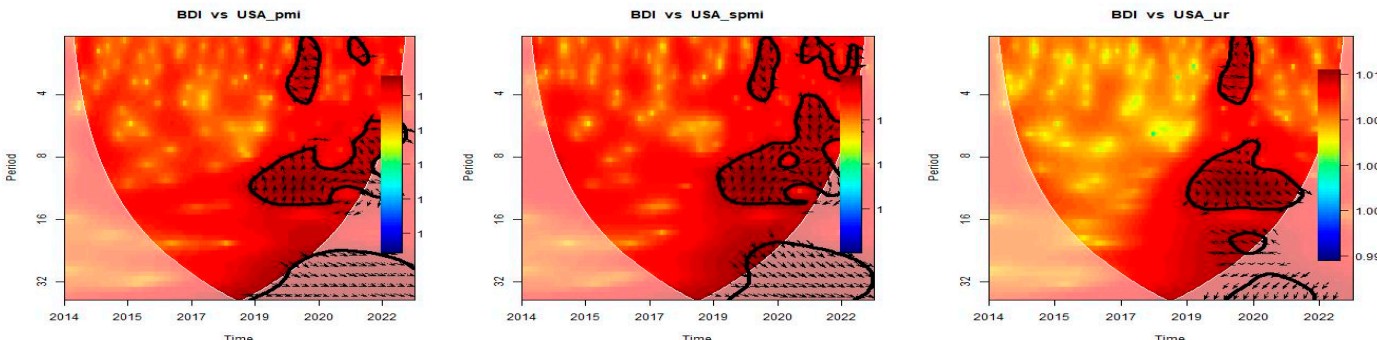

**Figure A3.** Pairwise wavelet analysis heatmaps for BDI vs. PMI, SPMI, and UR of six countries.

**Figure A4.** *Cont.*

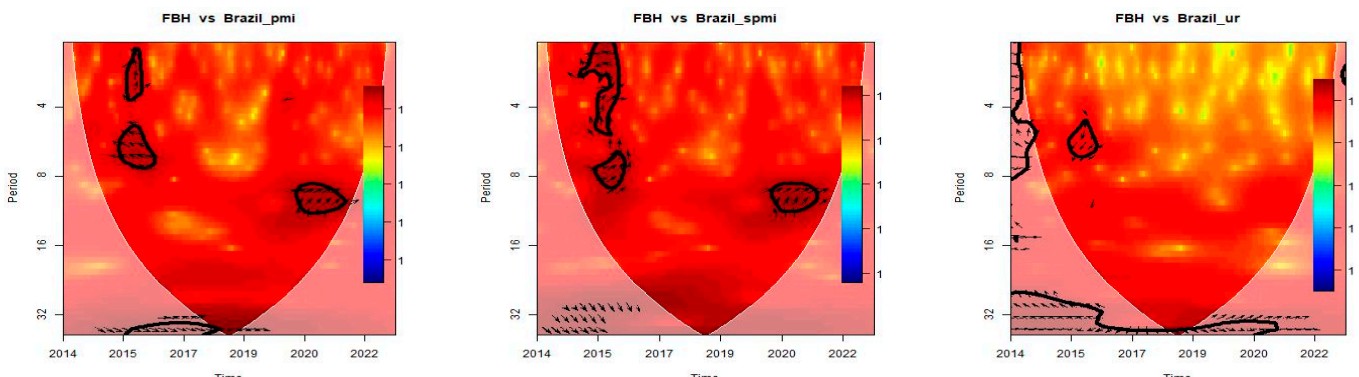

**Figure A4.** Pairwise wavelet analysis heatmaps for CEF vs. PMI, SPMI, and UR of six countries.

**Figure A5.** *Cont*.

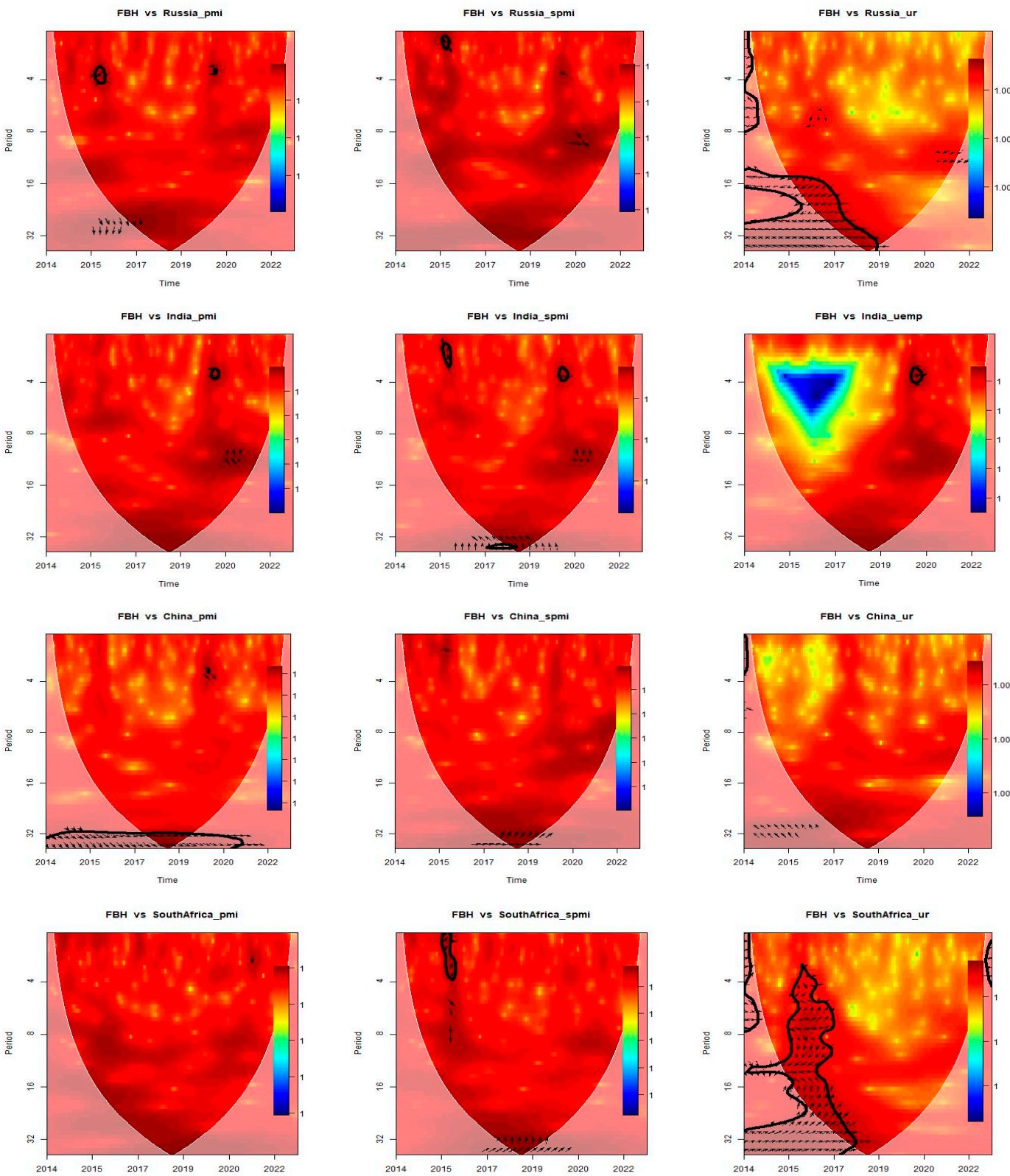

**Figure A5.** *Cont.*

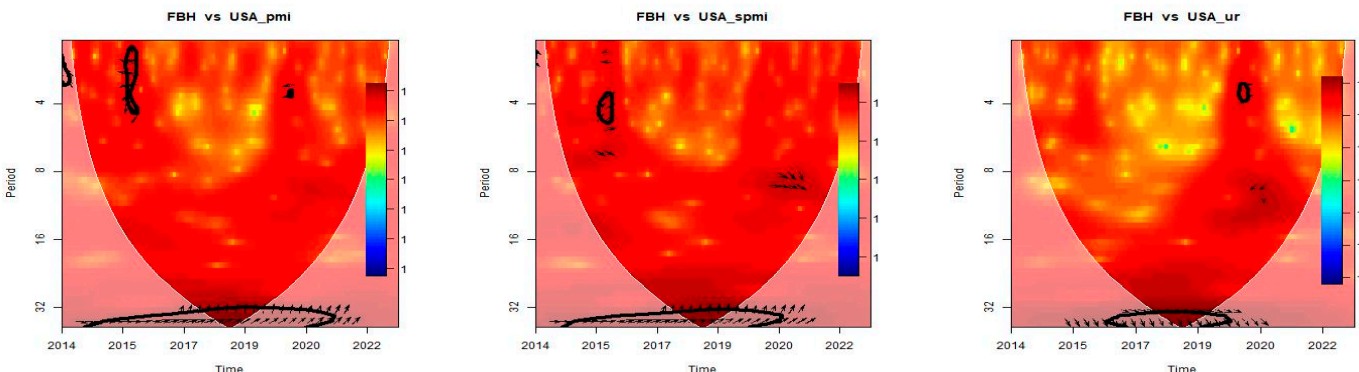

**Figure A5.** Pairwise wavelet analysis heatmaps for FBH vs. PMI, SPMI, and UR of six countries.

**Figure A6.** *Cont.*

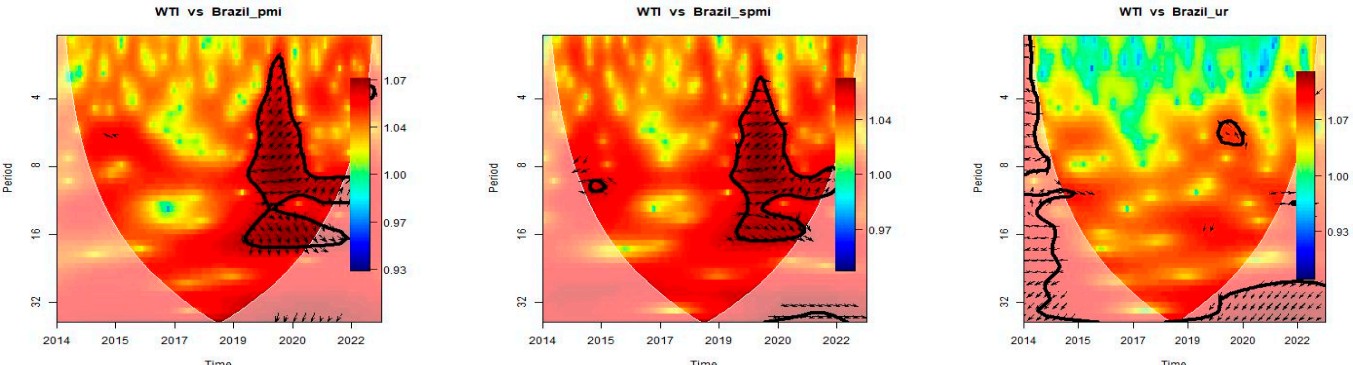

**Figure A6.** Pairwise wavelet analysis heatmaps for WEI vs. PMI, SPMI, and UR of six countries.

**Figure A7.** *Cont.*

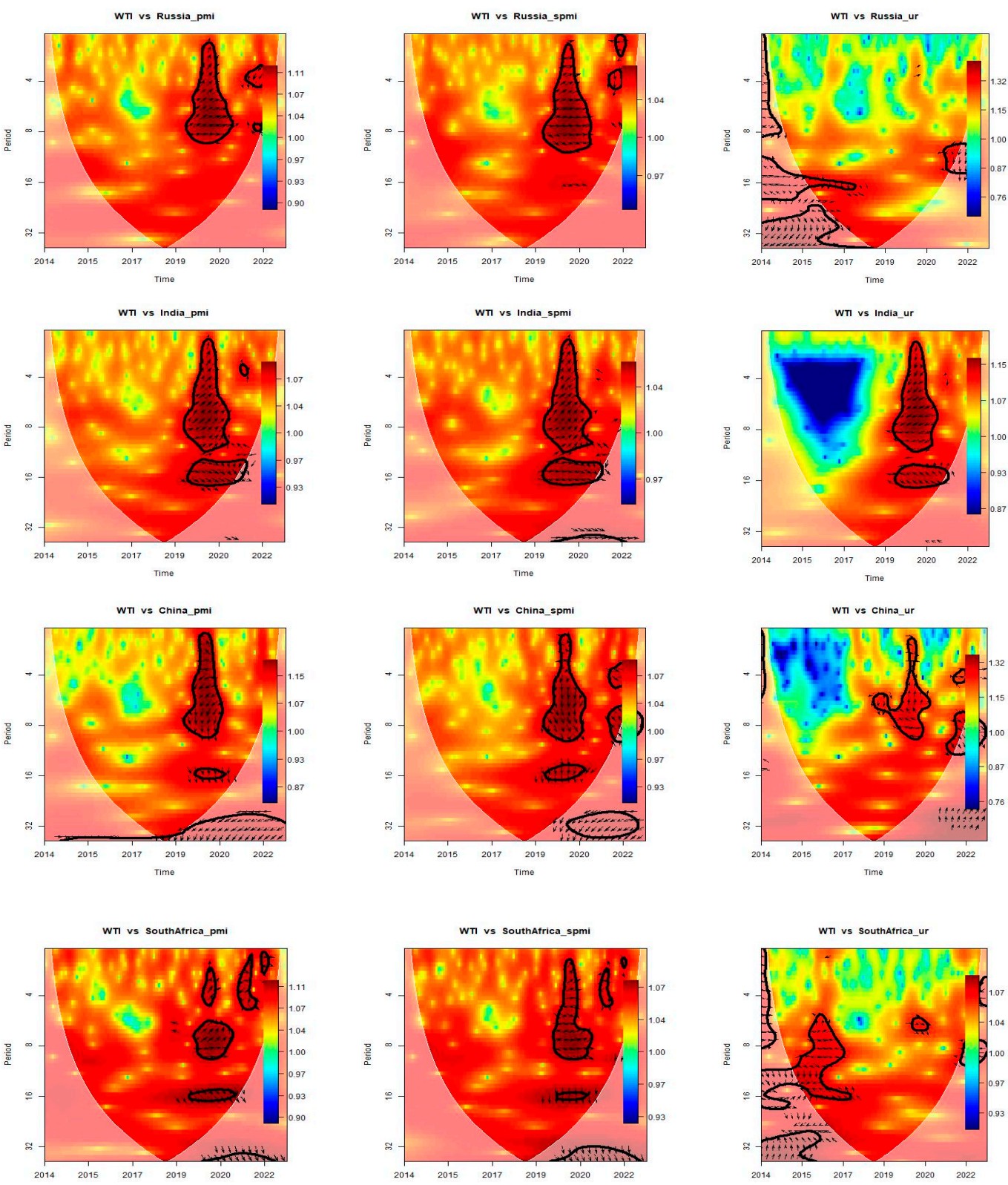

**Figure A7.** *Cont.*

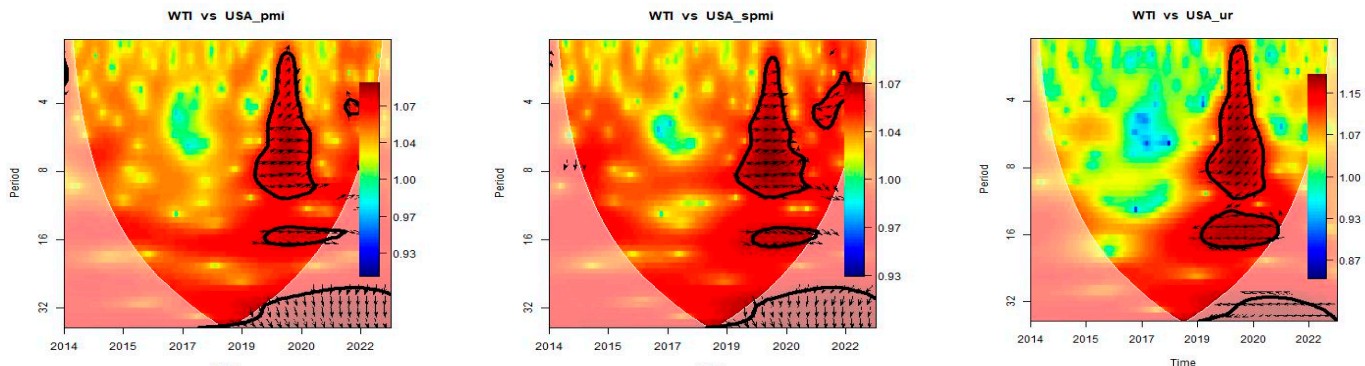

**Figure A7.** Pairwise wavelet analysis heatmaps for WTI vs. PMI, SPMI, and UR of six countries.

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
