# Peer review of "Assessing the Effect of the Magnitude of Spillovers on Global Supply Chains Using Quantile Vector Autoregressive and Wavelet Approaches"

_sustainability, doi:10.3390/su151914510_

Round 1

Reviewer 1 Report

Dear Authors, 

Thank you for the opportunity to get acquainted with an interesting article on the important topic of research on the temporal relationships of various economic and business variables with carbon emissions (also taking into account the business upheaval caused by the COVID-19 pandemic.

I like the use of sophisticated mathematical apparatus and new methods transferred from the area of telecommunication signal processing.

Before publishing, I recommend that the article be fine-tuned in terms of the following notes.

Best Regards and Good luck,

Reviewer

Comments:

  1. Line 144: "Cointegration between multiple variables is tested using the Jarque-Bera (JB)". The Jarque-Bera (JB) test is a statistical test used to test the normal distribution of data. Calculating the skewness and kurtosis of the data and the JB test statistics based on them does not make it possible to determine whether the linear combination of the two series is stationary. It is appropriate to explain why Jarque-Bera (JB) was used (a priori to determine normality) for cointegration testing (instead of the standard approach based on the properties of the eigenvalues of the cointegration vector matrices). The traditional cointegration methods are then written on lines 187-191. So, probably standard methods were used, and furthermore, the methods a priori applicable for the normality test were mixed in. It would be appropriate to make a correction here.
  2. It is nice to see the use of the Wavelet method in the field of time series analysis. This method, initially developed for signal analysis and image processing, is widely used in many fields, including time series analysis. The authors would add a formal (mathematical) description of the Wavelet method and state why it is possible to transfer this method to the field of time series.
  3. The authors should also add a mathematical description for the Augmented Dickey-Fuller (ADF) and Kwiatkowski-Phillips-Schmidt-Schin (KPSS) tests.
  4. Lines 187-191 have a large number of images. However, I cannot find a detailed explanation of these images. The authors should supplement the description of the figures.

Author Response

Authors’ Response to Reviewers’ comments/suggestions

Note: We refer to the reviews as coming from Reviewer #1, Reviewer #2, Reviewer #3, Reviewer #4, and Reviewer #5.  For each referee, we have addressed their comment/issue by giving a response in the revised paper and by providing an explanation here. For completeness, we reproduce the original referee comment/issue followed by our response.

For the most part, we agree with the remarks offered by the referees.  Where we don’t agree we state the reason for standing our ground.  It is our overall opinion that the referees’ comments were constructive and helped us produce a much better paper. All told, we have greatly expanded the organizational and methodological aspects of the paper and improved the paper’s readability. We think the revised paper will make a solid contribution to the existing literature. 

Our general Response to Reviewer #1:

The authors would like to thank Reviewer #1 for his/her constructive comments in the first version of the paper.

Comment #1

Line 144: "Cointegration between multiple variables is tested using the Jarque-Bera (JB)". The Jarque-Bera (JB) test is a statistical test used to test the normal distribution of data. Calculating the skewness and kurtosis of the data and the JB test statistics based on them does not make it possible to determine whether the linear combination of the two series is stationary. It is appropriate to explain why Jarque-Bera (JB) was used (a priori to determine normality) for cointegration testing (instead of the standard approach based on the properties of the eigenvalues of the cointegration vector matrices). The traditional cointegration methods are then written on lines 187-191. So, probably standard methods were used, and furthermore, the methods a priori applicable for the normality test were mixed in. It would be appropriate to make a correction here.

It is nice to see the use of the Wavelet method in the field of time series analysis. This method, initially developed for signal analysis and image processing, is widely used in many fields, including time series analysis. The authors would add a formal (mathematical) description of the Wavelet method and state why it is possible to transfer this method to the field of time series.

Response: Thank you for your valuable comment. In this revision, we added a mathematical description of the Wavelet method and explained why it can be used for the time series.

Comment #2

The authors should also add a mathematical description for the Augmented Dickey-Fuller (ADF) and Kwiatkowski-Phillips-Schmidt-Schin (KPSS) tests.

Response: Thank you for pointing this out too. In the revision, we also added the mathematical descriptions for the ADF and KPSS tests.

Comment #3

Lines 187-191 have a large number of images. However, I cannot find a detailed explanation of these images. The authors should supplement the description of the figures.

Response: Thank you for pointing out this need as well. More discussions about the figures are added into the relevant sections. 

Reviewer 2 Report

The article deals with the impact of the pandemic on global value chains. It contains a general review of the literature. Research results are insufficiently presented. The article requires a more detailed economic interpretation of the extent of spillovers to the global supply chains of selected countries. Citations do not comply with journal requirements.

Author Response

Authors’ Response to Reviewers’ comments/suggestions

Note: We refer to the reviews as coming from Reviewer #1, Reviewer #2, Reviewer #3, Reviewer #4, and Reviewer #5.  For each referee, we have addressed their comment/issue by giving a response in the revised paper and by providing an explanation here. For completeness, we reproduce the original referee comment/issue followed by our response.

For the most part, we agree with the remarks offered by the referees.  Where we don’t agree we state the reason for standing our ground.  It is our overall opinion that the referees’ comments were constructive and helped us produce a much better paper. All told, we have greatly expanded the organizational and methodological aspects of the paper and improved the paper’s readability. We think the revised paper will make a solid contribution to the existing literature. 

Our general Response to Reviewer #2

The authors would like to thank Reviewer #2 for his/her constructive comments. We have revised the paper along the lines suggested by Reviewer #2. The details of the changes to the paper follow here with Reviewer #2’s comments included for reference. Specific responses to key comments are given below:

Comment #1

The article deals with the impact of the pandemic on global value chains. It contains a general review of the literature. Research results are insufficiently presented. The article requires a more detailed economic interpretation of the extent of spillovers to the global supply chains of selected countries. Citations do not comply with journal requirements.

Response: Thank you for your valuable comments. In this revision, more discussions about the research results are added into the relevant sections as well as the economic interpretation of the spillovers to the supply chains as suggested. Referencing style is updated for consistency.

Reviewer 3 Report

Recommendations as to Publication:

I recommend the paper a Reject. The comments are as follows:

The authors explore the issues of the magnitude of spillovers on global supply chains. With a new conceptual econometric framework, the authors examine the dynamic connectedness among various economic indices in BRICS countries and U.S via a Quantile Vector Autoregressive model and wavelet approaches. After reviewing the paper, I found the paper was poorly accomplished. I recommend a rejection for the following reasons.

1.      The Introduction section introduces the industrial background and relevant issues, lacking an adequate introduction to the academic background.

2.      The authors should further demonstrate the research gap, motivation, and innovations, which are crucial for the paper.

3.      The literature review is not well presented. The authors should use subtitles to better categorize and organize the reviewed papers in various relevant research areas and accentuate the difference and relevance between the paper and the selected literature.

4.      As for the methodology, no details are provided to explicate why the Quantile Vector Autoregressive (QVAR) Method and Wavelet Method are used and how such methods are addressed to explore the research questions.

5.      The authors list various econometric indicators but do not explain the details of each indicator and why such indices are employed to construct the analysis framework.

6.      The research focuses on BRICS and U.S., which lacks generality for assessing the magnitude of spillovers on global supply chains.

7.      The analysis part is very weak. There is no analytical evidence to demonstrate how the results are obtained. No managerial insights are generated with convincing discussion.

8.      The conclusion part is weak, lacking real-world validation, research limitations, and future research suggestions.

The use of English is fine. 

Author Response

Authors’ Response to Reviewers’ comments/suggestions

Note: We refer to the reviews as coming from Reviewer #1, Reviewer #2, Reviewer #3, Reviewer #4, and Reviewer #5.  For each referee, we have addressed their comment/issue by giving a response in the revised paper and by providing an explanation here. For completeness, we reproduce the original referee comment/issue followed by our response.

For the most part, we agree with the remarks offered by the referees.  Where we don’t agree we state the reason for standing our ground.  It is our overall opinion that the referees’ comments were constructive and helped us produce a much better paper. All told, we have greatly expanded the organizational and methodological aspects of the paper and improved the paper’s readability. We think the revised paper will make a solid contribution to the existing literature. 

Our general Response to Reviewer #3

The authors would like to thank Reviewer #3 for his/her constructive comments and detailed instructions on revising the paper. Specific responses to key comments are given below:

Comment #1

The Introduction section introduces the industrial background and relevant issues, lacking an adequate introduction to the academic background.

Response: Thank you for your valuable comments. In this revision, a paragraph is added as an introduction to the academic background.

Comment #2

The authors should further demonstrate the research gap, motivation, and innovations, which are crucial for the paper.

Response: Thank you for your helpful suggestion. Additional statements are added to demonstrate the research gap, motivation, and contribution of the paper.

Comment #3

The literature review is not well presented. The authors should use subtitles to better categorize and organize the reviewed papers in various relevant research areas and accentuate the difference and relevance between the paper and the selected literature.

Response: Thank you for pointing this out. Literature review section is re-organized with subtitles as suggested wisely.

Comment #4

As for the methodology, no details are provided to explicate why the Quantile Vector Autoregressive (QVAR) Method and Wavelet Method are used and how such methods are addressed to explore the research questions.

Response: Thank you for point this out! In this revision, we added a mathematical description of the Wavelet method as well as QVAR method and explained why they can be used for to address our research question.

Comment #5

The authors list various econometric indicators but do not explain the details of each indicator and why such indices are employed to construct the analysis framework.

Response: Thank you for your helpful suggestion. In this revision, we provided explanations of each indicator and why they are used by providing an additional table.

Comment #6

The research focuses on BRICS and U.S., which lacks generality for assessing the magnitude of spillovers on global supply chains.

Response: This is one of the limitations that are added into the revision. Specifically, values of the target variables are not available for many countries. By focusing on the BRICS countries, we aim to build a framework and subsequently expand the scope with the inclusion of other countries. Moreover, on the brink of expansion discussions, BRICS countries are on the spotlight, thus we believe the timing of the study is very convenient.

Comment #7

The analysis part is very weak. There is no analytical evidence to demonstrate how the results are obtained. No managerial insights are generated with convincing discussion.

Response: Analysis part is improved by providing analytical insights to demonstrate how the results are obtained. A new sub-section is added to provide managerial insights.

Comment #8

The conclusion part is weak, lacking real-world validation, research limitations, and future research suggestions.

Response: The conclusion part is improved by adding statements about research limitations and future research directions as it was suggested.

Reviewer 4 Report

The overall research is less innovative. The data analysis part of the article needs to be improved. The results of the data analysis lacked corresponding managerial insights. And the structure of the article is not complete and lacks the necessary discussion.

Author Response

Authors’ Response to Reviewers’ comments/suggestions

Note: We refer to the reviews as coming from Reviewer #1, Reviewer #2, Reviewer #3, Reviewer #4, and Reviewer #5.  For each referee, we have addressed their comment/issue by giving a response in the revised paper and by providing an explanation here. For completeness, we reproduce the original referee comment/issue followed by our response.

For the most part, we agree with the remarks offered by the referees.  Where we don’t agree we state the reason for standing our ground.  It is our overall opinion that the referees’ comments were constructive and helped us produce a much better paper. All told, we have greatly expanded the organizational and methodological aspects of the paper and improved the paper’s readability. We think the revised paper will make a solid contribution to the existing literature. 

Our general Response to Reviewer #4:

The authors would like to thank Reviewer #2 for his constructive comments in the early version of the paper.

Comment #1

The overall research is less innovative. The data analysis part of the article needs to be improved. The results of the data analysis lacked corresponding managerial insights. And the structure of the article is not complete and lacks the necessary discussion.

Response: Thank you for pointing these out. Data analysis section is improved by providing an additional discussion of the extensive results obtained. Managerial insights are added as a new sub-section. The literature review section is reorganized around two sub-sections. The methodology section is revised by adding the mathematical formulations of QVAR and Wavelet analyses along with the reasoning behind their utilization.

Reviewer 5 Report

Referee Report for sustainability-2583325
Title: Assessing the Magnitude of Spillovers on Global Supply Chains using Quantile Vector Autoregressive and Wavelet Approaches

Overview

This paper uses QVAR approach to examine the negative impacts of the COVID-19 pandemic. After reviewing the paper in detail, I have the following comments:

Major Comments:

Comment 1: In the Introduction, the authors should highlight the importance of COVID-19 on supply chains.  

Comment 2: The main results of this paper should be summarized in the Introduction.

Comment 3: The following papers should be cited. For example, COVID-19 and E-Commerce Operations: Evidence from Alibaba; Filling a Theater During the COVID-19 Pandemic; Optimal innovation investment: The role of subsidy schemes and supply chain channel power structure;Effects of channel power structures on pricing and service provision decisions in a supply chain: A perspective of demand disruptions. Moreover, I suggest the authors add a table to highlight the contributions of the study.

Comment 4: The data source should be added. Some robust analysis is missing. The authors should extend the model to several directions.

Comment 5: The QVAR method should be clarified in section 3.

Comment 6: The impacts of COVID-19 on supply chain disruption is significant. The authors should point out how those results guide the firms choose suppliers, sales market and rebuild supply chain.

Referee Report for sustainability-2583325
Title: Assessing the Magnitude of Spillovers on Global Supply Chains using Quantile Vector Autoregressive and Wavelet Approaches

Overview

This paper uses QVAR approach to examine the negative impacts of the COVID-19 pandemic. After reviewing the paper in detail, I have the following comments:

Major Comments:

Comment 1: In the Introduction, the authors should highlight the importance of COVID-19 on supply chains.  

Comment 2: The main results of this paper should be summarized in the Introduction.

Comment 3: The following papers should be cited. For example, COVID-19 and E-Commerce Operations: Evidence from Alibaba; Filling a Theater During the COVID-19 Pandemic; Optimal innovation investment: The role of subsidy schemes and supply chain channel power structure;Effects of channel power structures on pricing and service provision decisions in a supply chain: A perspective of demand disruptions. Moreover, I suggest the authors add a table to highlight the contributions of the study.

Comment 4: The data source should be added. Some robust analysis is missing. The authors should extend the model to several directions.

Comment 5: The QVAR method should be clarified in section 3.

Comment 6: The impacts of COVID-19 on supply chain disruption is significant. The authors should point out how those results guide the firms choose suppliers, sales market and rebuild supply chain.

Author Response

Authors’ Response to Reviewers’ comments/suggestions

Note: We refer to the reviews as coming from Reviewer #1, Reviewer #2, Reviewer #3, Reviewer #4, and Reviewer #5.  For each referee, we have addressed their comment/issue by giving a response in the revised paper and by providing an explanation here. For completeness, we reproduce the original referee comment/issue followed by our response.

For the most part, we agree with the remarks offered by the referees.  Where we don’t agree we state the reason for standing our ground.  It is our overall opinion that the referees’ comments were constructive and helped us produce a much better paper. All told, we have greatly expanded the organizational and methodological aspects of the paper and improved the paper’s readability. We think the revised paper will make a solid contribution to the existing literature. 

Our general Response to Reviewer #5:

The authors would like to thank Reviewer #5 for his/her constructive and detailed comments. We have revised the paper along the lines suggested by Reviewer #5. Specific responses to key comments are given below:

Comment #1

In the Introduction, the authors should highlight the importance of COVID-19 on supply chains.  

Response: Thank you for your valuable comment. In this revision, the importance of COVID-19 on supply chains is further emphasized in the introduction section using the recently published industry reports.

Comment #2

The main results of this paper should be summarized in the Introduction.

Response: Thank you this suggestion. Main results are summarized in the Introduction as suggested.

Comment #3

The following papers should be cited. For example, COVID-19 and E-Commerce Operations: Evidence from Alibaba; Filling a Theater During the COVID-19 Pandemic; Optimal innovation investment: The role of subsidy schemes and supply chain channel power structure; Effects of channel power structures on pricing and service provision decisions in a supply chain: A perspective of demand disruptions. Moreover, I suggest the authors add a table to highlight the contributions of the study.

Response: Thank you providing these valuable resources! Suggest papers are cited accordingly in the literature review section.

Comment #4

The data source should be added. Some robust analysis is missing. The authors should extend the model to several directions.

Response: Thank you for suggesting the robustness analysis. In this review, it is added right before the Conclusions section. Data source is provided in the Data Availability Statement. It is aimed to extend the framework built in this study in other directions as wisely suggested.

Comment #5

The QVAR method should be clarified in section 3.

Response: Thank you for point out this need! In this revision, we added a mathematical description of the Wavelet method as well as QVAR method and explained why they can be used for to address our research question.

Comment #6

The impacts of COVID-19 on supply chain disruption is significant. The authors should point out how those results guide the firms choose suppliers, sales market and rebuild supply chain.

Response: Thank you for this very constructive feedback.  Although this study aims to help decision-makers understand the relationships between various factors and spillover effect, it is not intended to guide the firms in formulating specific strategies regarding their supply chains.  However, we will take this valuable comment as a suggestion for a future research direction.

Round 2

Reviewer 3 Report

I recommend the paper a Major Revision. The comments are as follows:

Compared to the previous, improvements can be observed in the resubmission. I have the following comments for the authors to improve the quality of the paper.

1.      The research questions are ambiguous. The research question depicted in the paper is more like a methodology, which should be improved.  

2.      As for the literature review, the authors should accentuate the difference and relevance between the paper and the selected literature to address the research significance of the paper.  

3.      As for the methodology, in addition to the mathematical description, there is no specific explanation to demonstrate why such approaches are appropriate and applicable to explore the research questions of this paper.

4.      As can be observed, a table is supplemented to summarize the econometric indicators. The authors should explain why such indices are employed to construct the analysis framework.

5.      The research only focuses on BRICS and the U.S., lacking generality for assessing the magnitude of spillovers on global supply chains, which does not accord with the paper title.

6.      The 4.2 Result is confusing and poorly organized. Where is Table A1 and A2? Where is Figure A3-A7? The authors should better code the numbers of tables and figures and better illustrate the results.

7.      As for the robustness analysis, the authors should explain how the figures are proposed.

8.      The managerial implication and conclusion still need further improvement to better address real-world validation in supply chain operations.

Moderate editing of English language required

Author Response

Our general Response to Reviewer #3:

We would like to thank Reviewer #3 for his/her constructive comments. We have revised the paper along the lines suggested. Specific responses to key comments are:

Comment #1

The research questions are ambiguous. The research question depicted in the paper is more like a methodology, which should be improved. 

Response: Thank you for your valuable observation.. We revised the research question as follows: What is the connection between the factors that expose the vulnerability of global supply chains and impact their performance?  To assess the vulnerability and spillover effect, two quantitative models are implemented to evaluate the connectedness relationship and spillover effects of shocks on time-series data.

Comment #2

As for the literature review, the authors should accentuate the difference and relevance between the paper and the selected literature to address the research significance of the paper.  

Response: Thank you for pointing this out too. An additional argument is added to stress what sets our paper apart from the existing ones and our contribution to the field. Below is the paragraph where we accentuated the relevance of our paper:

Although the supply chain literature on the COVID-19 pandemic started to build up sharply recently, mainly due to the limitations on data availability, there is a lack of quantitative models examining the relationships between key factors. This research fills this gap to enable decision-makers to implement informative decisions based on objective analytical results. Furthermore, available literature such as the above-mentioned papers investigating the interrelations among various factors does not account for the spillover effects of external shocks. Our paper contributes to the existing literature by assessing the connectedness of time-series data.

Comment #3

As for the methodology, in addition to the mathematical description, there is no specific explanation to demonstrate why such approaches are appropriate and applicable to explore the research questions of this paper.

Response: We hear you loud and clear. In this revision, we explained why the methodological approaches can be used for to address our research question. To assess the connectedness of time-series data, QVAR, and wavelet coherence methods can reveal the dynamic connectedness between variables and assess spillover effects of shock from the spatiotemporal perspective. Different quantiles in QVAR simulate different market conditions and Wavelet coherence method simulate the co-movement relationship. 

Comment #4

As can be observed, a table is supplemented to summarize the econometric indicators. The authors should explain why such indices are employed to construct the analysis framework.

Response: We gladly embraced your suggestion. In this revision, we provided explanations as to why these indicators are used to construct the analytical framework immediately after Table 1.

Comment #5

The research only focuses on BRICS and the U.S., lacking generality for assessing the magnitude of spillovers on global supply chains, which does not accord with the paper title.

Response: This is a good observation. However, it happens that those BRICS countries and the U.S. represent almost 75 percent of the combined worldwide trade (which goes hand-by-hand with supply chains) and enjoy a strategic political position in terms of their regional influence. Therefore, we firmly believe that the paper title rightly stands on its own.

Comment #6

The 4.2 Result is confusing and poorly organized. Where is Table A1 and A2? Where is Figure A3-A7? The authors should better code the numbers of tables and figures and better illustrate the results.

Response: Thank you for pointing this out. Tables and Figures coded as A# are located in the Appendix section. The location of such tables and figures are mentioned now where it applies.

Comment #7

As for the robustness analysis, the authors should explain how the figures are proposed.

Response: Thank you for helping clarify this. Figure 3 is constructed using the standard deviation of total spillover contribution from (to) other factors presented in Tables 3 and 4. This explanation is added to the section to clarify how Figure 3 in the Robustness Analysis section is constructed.

Comment #8

The managerial implication and conclusion still need further improvement to better address real-world validation in supply chain operations.

Response: You are indeed entitled to express your opinion. However, it seems we enter here into the realm of an apparent subjectivity as to whether or not our elaboration on the managerial implication and conclusion adequately conveys what is extractable from the research and results obtained. We think the current title is purposedly catchy enough to entice the interest of other researchers in the field.

Reviewer 4 Report

The author has already made the corresponding changes based on my comments

Author Response

Comment: The author has already made the corresponding changes based on my comments

We would like to thank Reviewer #4 for his/her approval.

Reviewer 5 Report

Dear Editors,

The authors have addressed my all concerns.

Thanks

Dear Editors,

The authors have addressed my all concerns.

Thanks

Author Response

Comment: The authors have addressed my all concerns.

We would like to thank Reviewer #5 for his/her approval.

Round 3

Reviewer 3 Report

The revision and improvement can be observed in the new resubmission. 

The use of English is fine.